# Feature-specific prediction errors and surprise across macaque fronto-striatal circuits

Mariann Oemisch[1,2], Stephanie Westendorff[1,3], Marzyeh Azimi[1], Seyed Alireza Hassani[1,4], Salva Ardid [5], Paul Tiesinga[6] & Thilo Womelsdorf [1,4]

To adjust expectations efficiently, prediction errors need to be associated with the precise features that gave rise to the unexpected outcome, but this credit assignment may be problematic if stimuli differ on multiple dimensions and it is ambiguous which feature dimension caused the outcome. Here, we report a potential solution: neurons in four recorded areas of the anterior fronto-striatal networks encode prediction errors that are specific to feature values of different dimensions of attended multidimensional stimuli. The most ubiquitous prediction error occurred for the reward-relevant dimension. Feature-specific prediction error signals a) emerge on average shortly after non-specific prediction error signals, b) arise earliest in the anterior cingulate cortex and later in dorsolateral prefrontal cortex, caudate and ventral striatum, and c) contribute to feature-based stimulus selection after learning. Thus, a widely-distributed feature-specific eligibility trace may be used to update synaptic weights for improved feature-based attention.

[1] Department of Biology, Centre for Vision Research, York University, 4700 Keele Street, Toronto, ON M6J 1P3, Canada. [2] Department of Neuroscience, Yale University School of Medicine, New Haven, CT 06510, USA. [3] Institute of Neurobiology, University of Tübingen, Tübingen 72076, Germany. [4] Department of Psychology, Vanderbilt University, Nashville, TN 37240, USA. [5] Department of Mathematics and Statistics, Boston University, Boston, MA 02215, USA. [6] Donders Institute for Brain, Cognition and Behaviour, Radboud University Nijmegen, Nijmegen 6525 EN, Netherlands. Correspondence and requests for materials should be addressed to M.O. (email: mariann.oemisch@yale.edu) or to T.W. (email: thilo.womelsdorf@vanderbilt.edu)

When faced with novel objects we learn about the relevance of their dimensions (e.g., color) and features (e.g., red), by estimating feature values and improving this estimate through trial and error learning[1,2]. Computationally, this can be achieved by calculating how unexpected an experienced outcome is, and updating value estimates in proportion to this unexpectedness[3,4]. In typical reinforcement learning (RL) models, the unexpectedness is calculated as prediction error between predicted value and experienced outcome[5].

A prominent hypothesis suggests that the degree of unexpectedness is guiding the subject's future attention toward the specific features that gave rise to an unexpected outcome[6,7]. The biasing of attention to those features whose reward prediction is most strongly violated can optimize sampling of visual information[8,9]. Recent evidence supports this view by showing that attention biases closely follow the distribution of feature values[3,4,10,11]. Instead of attending all dimensions of a stimulus equally, prioritizing dimensions that are most reward predictive, dramatically enhances learning speed when stimuli are composed of multiple dimensions[1,12]. These findings predict that brain circuits combine information about the occurrence of a prediction error with information about the specific stimulus feature of the relevant dimension that should be attended in future trials[13]. However, it is unknown how this combination of prediction error information and feature-based attention is realized in brain circuits.

Here, we address this question by quantifying how prediction errors are encoded for task-relevant features within four areas of the medial and lateral anterior fronto-striatal loops[14]. We asked (1) whether prediction error signals in these regions are informative of the specific features that were chosen (upwards motion, color red, etc.), and (2) whether such feature-specific encoding of prediction errors occurs more commonly for the reward-relevant dimension as opposed to reward-irrelevant dimensions. We did so using a task that employed stimuli that could be characterized by multiple dimensions (color, location, and motion), of which however only one was linked to reward outcome across trials (color). Feature values within this reward-relevant dimension were then reversed, akin to intradimensional shifts in the set-shifting literature (e.g.,[15]).

Learning in such a task might be accomplished with a localized, general prediction error in the ventral striatum (VS) that is then broadcasted to prefrontal cortex where it modifies the activity of feature-selective neurons[13]. This view is supported by mostly human functional magnetic resonance imaging findings that single out the striatum as core region to encode prediction errors[16], and the lateral prefrontal cortex to encode feature-based top-down signals[3,17,18] together with prediction errors[13]. In contrast to such a scenario, neurons encoding prediction errors might be distributed widely and carry explicit feature-choice information in multiple areas[19]. Activity of such neurons could serve as a feature-specific eligibility trace[20], orchestrated across the recurrent fronto-striatal loops. Such a distributed, feature-specific eligibility trace is predicted by network models that learn relevant features by using attentional feedback signals to label synapses of those neurons that contributed to the feature-specific reward prediction itself[21,22].

Here, we found support for distributed feature-specific encoding of prediction errors across the anterior cingulate and prefrontal cortex, as well as the connected striatal projection regions in VS and caudate head. The neural feature-specific encoding of prediction errors emerged on average after the

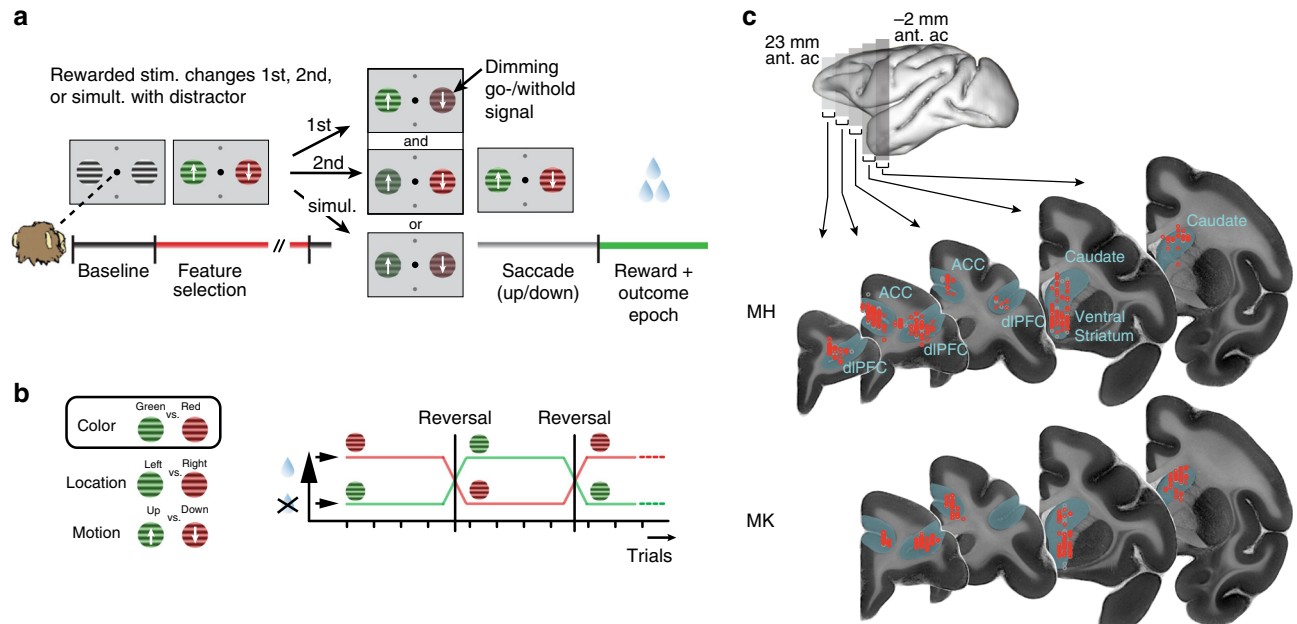

**Fig. 1** Feature-based reversal learning task and anatomical recording locations. **a** Animals are presented with two black/white stimulus gratings to the left and right of a central fixation point. The stimulus gratings then become colored and start moving in opposite directions. Dimming of the stimuli served as Choice/Go signal. At the time of the dimming of the target stimulus the animals had to indicate the motion direction of the target stimulus by making a corresponding up or downward saccade in order to receive a liquid reward. Dimming of the target stimulus occurred either before, after or at the same time as the dimming of the distractor stimulus. **b** Left: Three features characterize each stimulus—color, location, and motion direction. Only the color feature is directly linked to reward outcome. The task is a deterministic reversal learning task, whereby only one color at a time is rewarded. Right: This reward contingency switches repeatedly and unannounced in a block-design fashion. **c** Illustration of recording locations relative to stereotaxic zero for monkey H (top) and monkey K (bottom). Neuron locations are collapsed across 5 mm coronal slices indicated by the gray bars on the brain on top. Red circles represent neurons that encoded a feature-specific prediction error, gray circles represent neurons that did not. Ant. ac. refers to anterior of anterior commissure. Imaging data provided by the Duke Center for in vivo microscopy[73,74]

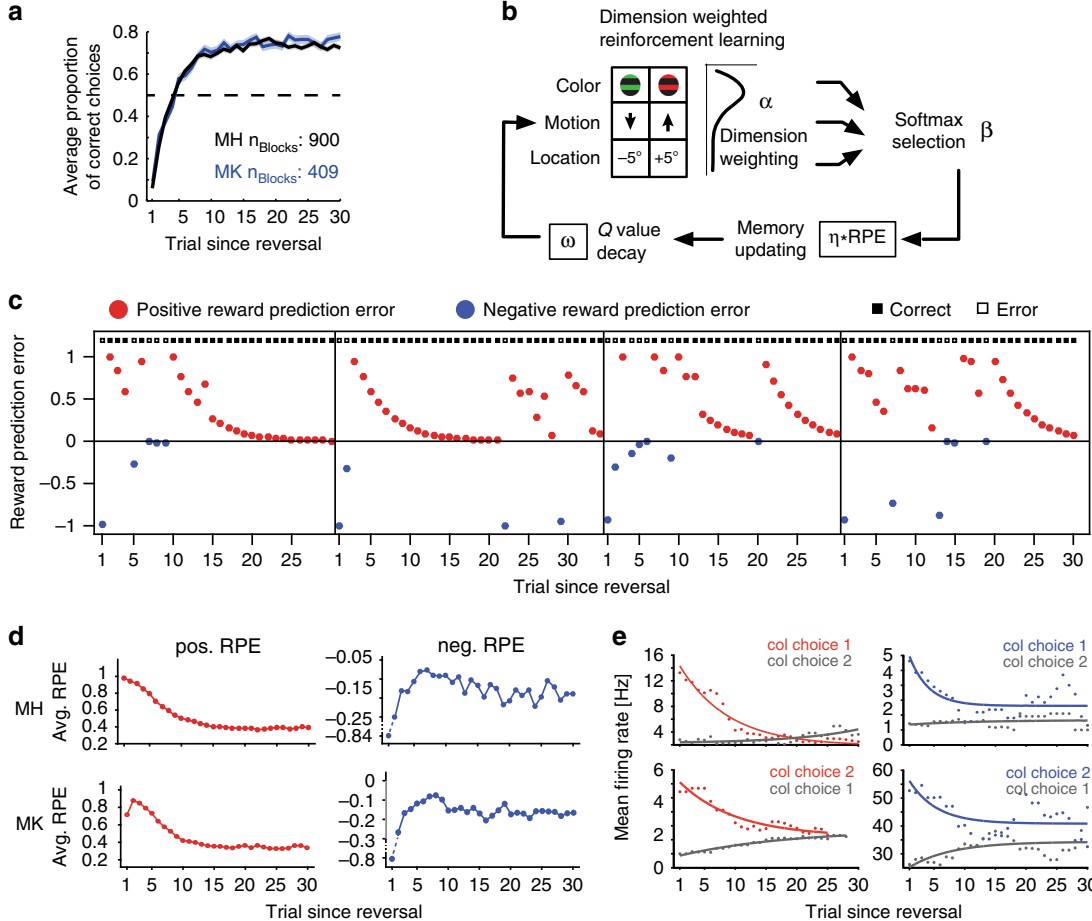

**Fig. 2** Task performance and reinforcement learning model-derived RPEs. **a** Average proportion of correct choices relative to the reversal for monkey H (gray) and monkey K (blue). Shaded error bars represent standard error of the mean (SEM). Numbers of blocks included in the analysis are indicated for both monkeys. **b** Dimension weighted reinforcement learning model with main parameters alpha, beta, omega, and eta ($\alpha$, $\beta$, $\omega$, and $\eta$) for feature weighting, selection noise, decay rate, and learning rate, respectively. **c** Model-derived reward prediction errors for four example reversal blocks. Positive RPEs are indicated in red, negative RPEs are indicated in blue. Small black filled and white filled squares at the top of each graph indicate correct and error choices, respectively. **d** Average positive (left) and negative (right) model-derived RPEs across the first 30 trials of a reversal block for monkey H (top) and monkey K (bottom). **e** Average firing rates of four example neurons in the outcome epoch following color 1 and color 2 choices across trials in a reversal block, in correct trials (left two neurons) and in error trials (right two neurons). These neurons were later identified to encode color-specific positive and negative RPEs, respectively. For visualization purposes only, exponential functions were fit to the data points

encoding of nonspecific prediction errors and was conveyed by neurons that showed stronger attentional selection signals in subsequent trials, thus potentially contributing to improved learning and visual selection.

## Results

**Behavior.** Monkeys performed a reversal learning task which presented two peripheral stimuli with different colors and motion directions (Fig. 1a). Over sequences of 30 or more trials, one of two colors was associated with reward outcomes (juice drops), while features of other stimulus dimensions (left vs. right stimulus location, up vs. downward motion direction) were not linked to reward (Fig. 1b). To obtain reward, the animals had to wait for a Go-signal (dimming of the stimuli) and make a saccade in the motion direction of the stimulus whose color matched the reward-associated color. The reward schedule in this task was deterministic. This task required (1) feature-based attentional selection of one stimulus based on a reward-associated color, (2) to use the motion direction of the attended stimulus to program a saccadic response, and (3) to make a response only when the attended stimulus dimmed. Therefore, stimulus location (for

spatial attention) and stimulus motion direction (for action planning) were task-relevant on a trial-by-trial basis, while only stimulus color was linked to reward across trials.

Both monkeys learned this feature-specific credit assignment and adjusted their attention bias to the reward-associated color after uncued reversals (Fig. 2a). As estimated with an ideal observer statistic[23,24] (Supplementary methods), monkeys H/K successfully learned on average $83 \pm 2/91 \pm 2\%$ of blocks, whereby learning occurred on average within $17.5 \pm 0.5/16.5 \pm 0.6$ trials following a reversal. Monkeys H/K performed on average $8.7 \pm 0.3/8.9 \pm 0.3$ reversal blocks per recording session with average block lengths of $45 \pm 0.7/43 \pm 0.8$ trials (median: 37/36).

**Encoding of outcome and feature-specific prediction errors.** We recorded 1960 units in two monkeys with 690 units in ACC (monkey H/K: 405/285), 524 units in dlPFC (monkey H/K: 316/208), 449 units in caudate nucleus (CD; monkey H/K: 234/215), and 297 units in VS (monkey H/K: 163/134) (Fig. 1c). In total, 71%/78% of neurons in monkey H/K met the criteria for analysis (see Methods). Among these neurons, 38% encoded outcome (rewarded versus unrewarded), ranging between 27 and 53%

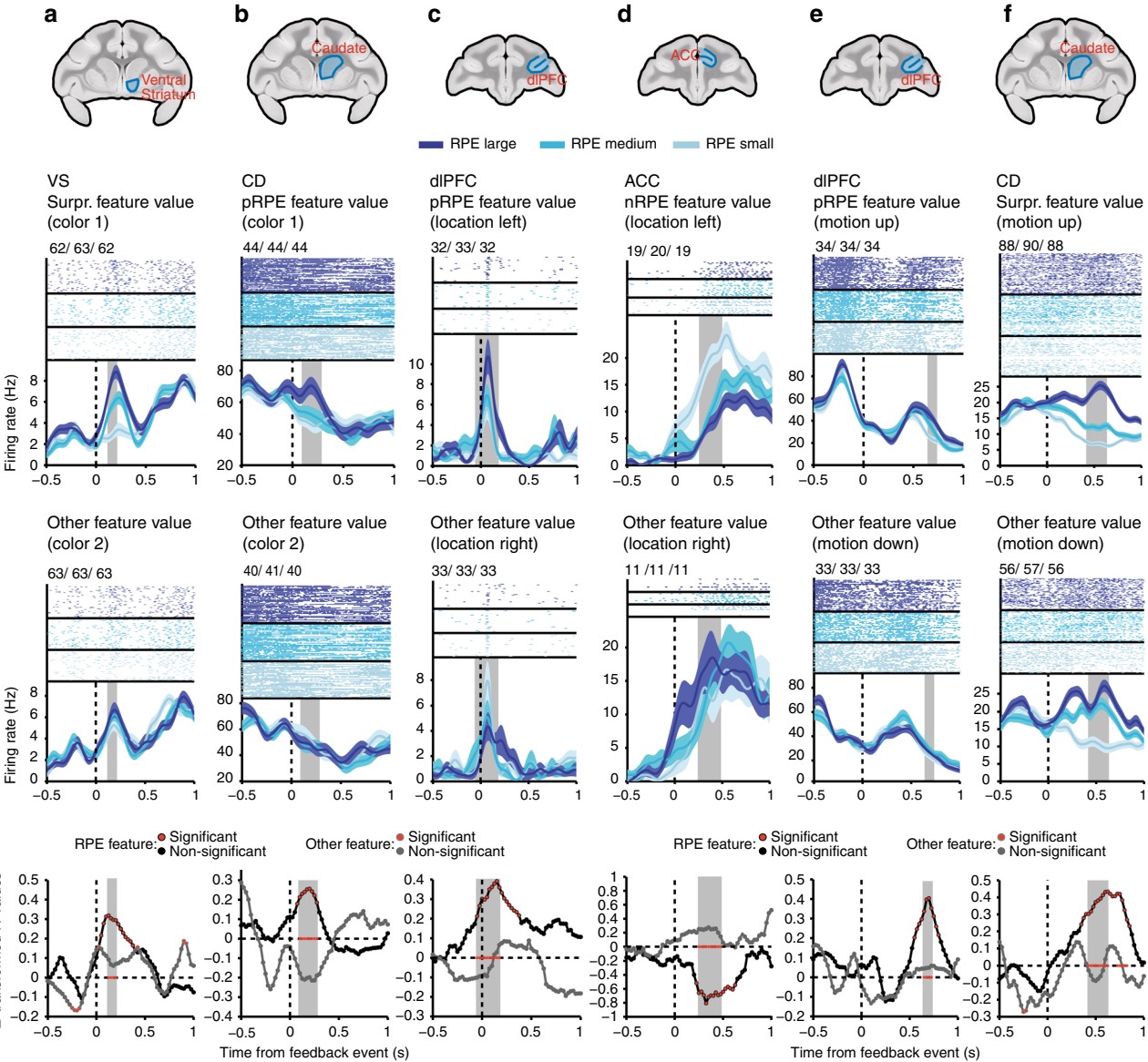

**Fig. 3** Example neurons encoding feature-specific RPE signals. For each of six example neurons (**a–f**), the spike rasters and spike-density functions are displayed (for visualization purposes only) for prediction error values of three different magnitudes (trials evenly split into RPE large, RPE medium, and RPE small), for the feature (e.g., color 1) for which an RPE was encoded (preferred RPE feature, top row), and for the feature for which an RPE was less or not encoded (e.g., color 2, nonpreferred RPE feature, and middle row). The bottom most row displays the z-transformed R values of the correlation between spike rate and RPE for the two feature values above (solely this last row displays the statistical analyses performed). Black (nonsignificant) or red filled (significant) circles represent z-transformed R values of the correlation between spike rate and RPE for preferred RPE feature trials. Gray (nonsignificant) or red filled (significant) outlines represent z-transformed R values of the correlation between spike rate and RPE for nonpreferred RPE feature trials (Spearman correlation). Red stars indicate those time bins for which the R values between the two feature values differed significantly (Z-test, see Methods, Eq. ( 1)). Gray transparent bars in all plots indicate the time window of RPE encoding. Small numbers at the top left of each subplot containing spike rasters indicate the numbers of trials that were included in the RPE large/RPE medium/RPE small groups, respectively. The title above each column of figures indicates the area that neuron was recorded from as well as the type of feature and RPE signal encoded by that neuron. Anatomical images at the top-most additionally illustrate the recording locations. Shaded error bars represent SEM. Imaging data provided by the Duke Center for in vivo microscopy[73,74]

across brain areas, as determined using linear regression analysis in the 0.1–0.7 s following feedback onset (Supplementary methods and Figs. 2a, 3a, b).

To discern trial-by-trial encoding of reward prediction errors (RPEs), we first evaluated various RL models using RPEs to account for feature-based learning performance[4,10] (Supplementary methods). The models used RPEs to update values either (1) for all stimulus features independently, (2) for all stimulus features weighted by their dimensional relevance, or (3) for only

the different colors. We found that the monkeys' learning performance was best explained by an RL model that learns to weight values of specific features by their dimension, i.e., that learned a location, motion direction and color weight, and additionally decayed feature-specific values of the nonchosen stimulus. These findings corroborate previous studies[4,10], suggesting that both dimension and feature information contribute to learning performance (Fig. 2b, Supplementary Fig. 1, Supplementary Methods). We used this RL model to generate

trial-by-trial estimates of RPEs (see Fig. 2c) for correlating firing rates during the −0.5 to 1.5 s outcome epoch of each trial time-resolved in 200 ms windows shifted by 50 ms (Fig. 2d, e).

Neurons were classified as encoding RPEs when their firing rates were significantly correlated with model RPEs in a minimum of four consecutive time windows (≥0.1 s) (Spearman correlation, see Methods). Negative RPE (nRPE encoding neurons were indexed as those neurons that increased their firing rates with more negative nRPE values[25,26]. By definition, positive RPEs (pRPE) occurred solely on correct trials, nRPEs occurred solely on error trials. For monkeys H/K firing rates correlated significantly with pRPE in 21/22% of neurons, with nRPE in 14/10% of neurons, and for 24/24% of neurons with the unsigned RPE that indexes surprise (e.g.,[27]). RPE correlations of firing were evident in all areas and monkeys (Supplementary Fig. 2b). RPE's were computed as $RPE = R − V$, where $R$ denotes reward (always 1 or 0), and $V$ denotes expected value of the chosen stimulus. This formulation also shows that RPE is positively correlated with reward ($R$, always 0/1), and anti-correlated ($−V$) with the value of the chosen stimulus prior to the time of reward throughout task performance.

We hypothesized that to effectively use RPE information to adjust feature-based attention, (1) neurons may selectively encode an RPE for one of two chosen feature values (e.g., for color 1 but not color 2), and (2) across neurons, such RPEs are not equally

encoded for all stimulus dimensions, but selectively for the task-relevant dimension (color vs. location/motion). Consistent with the first hypothesis, we found evidence for feature-selective RPE encoding in multiple neurons (Fig. 2e, Fig. 3). For instance, the VS neuron in Fig. 3a responded from weak to strong when the absolute RPEs (surprise) were low to high when color 1 was chosen (upper panel), but showed no firing modulation with RPEs when color 2 was chosen (middle panel), resulting in a significant color-selective RPE × firing rate correlation in the 100–200 ms following reward onset (bottom panel). Similar examples were evident for other feature dimensions. The ACC neuron in Fig. 3d showed stronger firing with more negative nRPE only when the selected stimulus was located on the left (top panel), but not when it was located on the right (middle panel). Finally, the dlPFC neuron in Fig. 3e fired stronger with larger pRPEs when the motion of the chosen stimulus was upwards (top panel), but not when it was downwards (middle panel).

Overall, we found that 53.1% of neurons (52.7%/53.6% in monkey H/K) across the fronto-striatal areas tested encoded feature-specific positive, negative, and unsigned (surprise) prediction error signals (for their anatomical reconstruction see Supplementary Figs. 4, 5). Indeed, most neurons (80%) that were initially identified to encode non-specific RPEs (see above, Supplementary Fig. 2b) encoded an RPE dependent on the

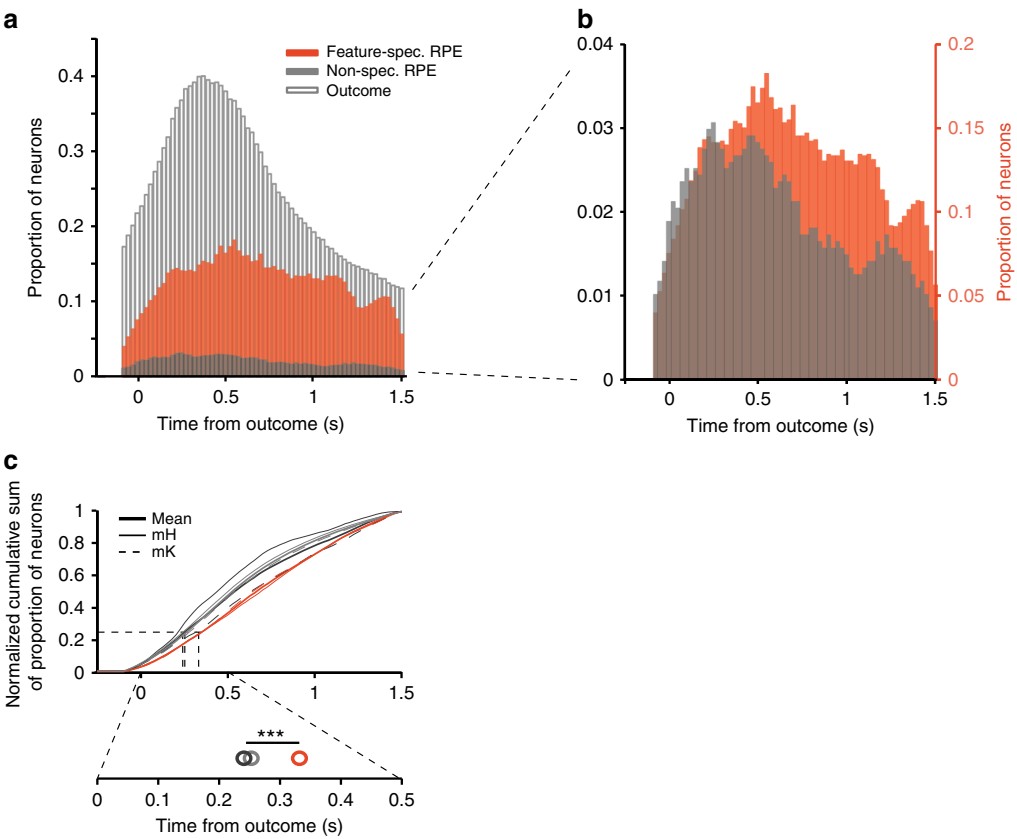

**Fig. 4** Temporal profile of feature-specific RPE, nonspecific RPE and outcome signals. **a** Histogram of the proportion of units encoding feature-specific RPEs, nonspecific RPEs and outcome in time, combined across both monkeys. For each neuron, all time bins for which an RPE/outcome was encoded are included ($n_{\text{feat-spec}} = 774$; $n_{\text{non-spec}} = 167$). **b** Displays the same as (**a**) for nonspecific and feature-specific RPE encoding, but zoomed in and scaled similarly to enhance comparison of the two RPE types. **c** Normalized cumulative sums of the histograms in (**a**). Top: Thick lines represent the mean across both monkeys, while thin continuous lines represent cumulative sums of monkey H, and thin dotted lines represent cumulative sums of monkey K. The cumulative sum of feature-specific RPEs differed significantly from those of nonspecific RPEs and outcome (Kolmogorov–Smirnoff test, Bonferroni–Holm multiple-comparison correction; both $p < .001$). Bottom: Magnification of the cumulative sums around the 25% window. Open circles represent the time points at which 25% of the respective signal is encoded. The horizontal bar with three asterisks indicates that the 25% time point of feature-specific RPEs differs significantly from those of nonspecific RPEs and outcome (randomization procedure, both $p < 0.001$)

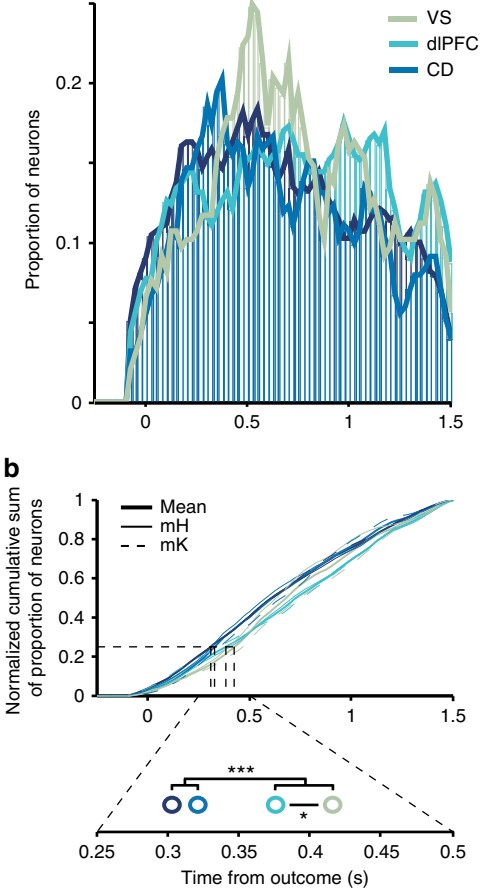

**Fig. 5** Latency comparison of feature-specific RPE encoding across areas. **a** Histogram of the proportion of feature-specific RPE encoding units in ACC, VS, dlPFC, and CD combined in time across both monkeys. For each neuron, all time bins for which an RPE was encoded are included ($n_{ACC}$ = 256; $n_{VS}$ = 132; $n_{dlPFC}$ = 234; $n_{CD}$ = 152). To enhance visualization of the four histograms lines representing the outlines of each histogram are added. **b** Normalized cumulative sums of the histograms in (**a**). Top: Thick lines represent the mean across both monkeys, while thin continuous lines represent cumulative sums of monkey H, and thin dotted lines represent cumulative sums of monkey K. The cumulative sums of all areas except for ACC and CD differed significantly from each other (Kolmogorov–Smirnoff test, Bonferroni–Holm multiple-comparison correction; $p_{ACC-CD}$ = 0.128, all other $p$ < 0.01). Bottom: Magnification of the cumulative sums around the 25% window. Open circles represent the time points at which 25% of feature-specific RPEs are encoded in the four areas. One asterisk indicates $p$ < 0.05; three asterisks indicate $p$ < 0.001 (randomization procedure). Hedges' $g$ effect sizes for the latency differences: $d_{ACC-dlPFC}$ = −0.18, $d_{ACC-CD}$ = −0.02, $d_{ACC-VS}$ = −0.16, $d_{dlPFC-CD}$ = 0.17, $d_{dlPFC-VS}$ = 0.03, $d_{CD-VS}$ = −0.14

feature of the chosen stimulus, i.e., they encoded feature-specific RPEs (80.6%/79.5% in monkey H/K).

**Feature-specific RPEs emerge later than nonspecific RPEs.** Feature-specific RPE signals might arise from neurons that initially encode the occurrence of a nonspecific RPE by combining RPE with chosen feature information over time. This scenario predicts a slower time course of more specific information about the source of the error[28]. We tested this by comparing for each neuron the time windows in which it significantly encoded a feature-specific RPE, nonspecific RPE, or outcome per se for a minimum of four consecutive time bins (≥0.1 s). For

this analysis, we collapsed across neurons encoding color/location/motion-specific RPEs and grouped neurons into those encoding either feature-specific, or nonspecific RPEs (see Methods). We found that on average feature-specific RPE encoding emerged later than nonspecific RPE encoding as indexed by a shallower slope of the temporal cumulation of the proportion of significant RPE encoding neurons (Kolmogorov–Smirnoff test, Bonferroni–Holm corrected: $p_{feat-non}$ < .001; $D_{feat-non}$ = 0.10, $n_{feat-spec}$ = 774; $n_{non-spec}$ = 167) (Fig. 4). This was additionally confirmed using nonparametric statistics (Rank-sum test, Bonferroni–Holm corrected, $W$ = 8.4 × 10⁶, $p$ < .0001, Hedges' $g$ = −0.21). The effect was partly driven by feature-specific RPE encoding neurons showing a continued increase and remaining at a higher plateau level for a longer duration than nonspecific RPE signals (Fig. 4a, b, Figure S6a). Thus, while some individual neurons did show early feature-selective RPE encoding, at the population level, the set of neurons encoding feature-specific error information showed a slower time course, which was sustained at a higher level compared to nonspecific error information. Feature-specific RPEs on average also emerged significantly later than encoding of rewarded/nonrewarded outcome (Kolmogorov–Smirnoff test, Bonferroni–Holm corrected, $p_{feat-out}$ < 0.001, $D_{feat-non}$ = 0.12, Hedges' $g$ = 0.26, $n_{out}$ = 702), while nonspecific RPEs were encoded at a similar time as outcome ($p_{non-out}$ = 0.089, $D_{feat-non}$ = 0.03, Hedges' $g$ = 0.05). Supporting these latency findings, we found that 25% of outcome encoding occurred at 268 ms after feedback onset, 25% of nonspecific RPE encoding occurred at 255 ms, and 25% of feature-specific RPE encoding occurred at 355 ms (randomization statistic: $p_{feat-non}$ < 0.001; $p_{feat-out}$ < 0.001; $p_{non-out}$ = 0.27; Fig. 4c). These results were robust to the statistical criterion for identifying RPE encoding neurons (Supplementary Fig. 6a).

We next asked when feature-specific RPE encoding emerged in each of the four brain areas. Using the same latency measures as above, we found that the rise of neurons with significant feature-specific RPE differed significantly between all areas, except for ACC and CD (Kolmogorov–Smirnoff test, Bonferroni–Holm corrected: $p_{ACC-dlPFC}$ < 0.001, $D_{ACC-dlPFC}$ = 0.09; $p_{ACC-CD}$ = 0.128, $D_{ACC-CD}$ = 0.03; $p_{ACC-VS}$ < 0.001, $D_{ACC-VS}$ = 0.09; $p_{dlPFC-CD}$ < 0.001, $D_{dlPFC-CD}$ = 0.09; $p_{dlPFC-VS}$ = 0.006, $D_{dlPFC-vs}$ = 0.05; $p_{CD-VS}$ < 0.001, $D_{CD-VS}$ = 0.09) (Fig. 5a, b). Feature-specific RPE signals emerged earliest in the ACC (310 ms) and CD (330 ms), followed by dlPFC (385 ms), followed by VS (428 ms) (randomization statistic: $p_{ACC-dlPFC}$ < 0.001; $p_{ACC-CD}$ = 0.136; $p_{ACC-VS}$ < 0.001; $p_{dlPFC-CD}$ < 0.001; $p_{dlPFC-VS}$ = 0.018; $p_{CD-VS}$ < 0.001; Fig. 5b bottom). Using a nonparametric measure confirmed these results, except for no significant latency difference between dlPFC and VS (Rank-sum test, Bonferroni–Holm corrected: $p_{ACC-dlPFC}$ < 0.001, $W_{ACC-dlPFC}$ = 11 × 10⁶; $p_{ACC-CD}$ = 0.514, $W_{ACC-CD}$ = 9 × 10⁶; $p_{ACC-VS}$ < 0.001, $W_{ACC-VS}$ = 8.5 × 10⁶; $p_{dlPFC-CD}$ < 0.001, $W_{dlPFC-CD}$ = 8.9 × 10⁶; $p_{dlPFC-VS}$ = 0.345, $W_{dlPFC-VS}$ = 8.3 × 10⁶; $p_{CD-VS}$ < 0.001, $W_{CD-VS}$ = 3.2 × 10⁶). These results were robust to the statistical criterion for identifying significant RPE encoding (Supplementary Fig. 4b).

**Feature-tuning of RPEs.** To update the attentional set to the goal-relevant color during reversal learning, neurons should preferentially encode prediction errors for the reward-relevant color dimension as opposed to the motion and location dimension that were task relevant only for completing individual trials[29]. Consistent with this rationale, we found nRPEs and pRPEs were encoded more often for the color dimension, than for location or motion (one-sided bootstrap CI: $p$ ≤ 0.05; Fig. 6a, d, respectively). Neurons encoding color-specific nRPEs were more prevalent in ACC, VS, and dlPFC (one-sided bootstrap CI: $p$ ≤ 0.05, Fig. 6b). We

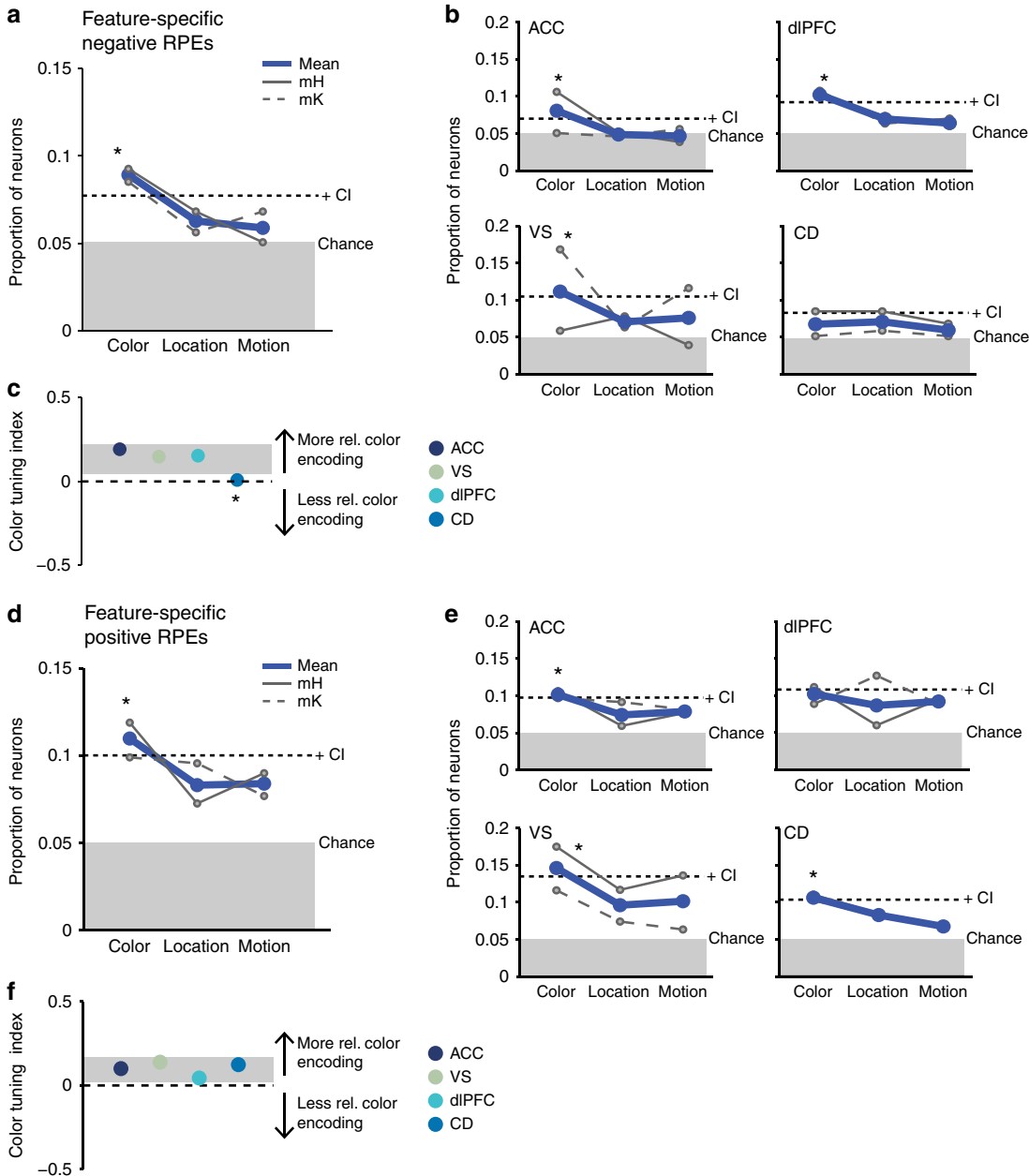

**Fig. 6** Prevalence of feature-specific negative and positive RPE encoding. Shown are proportions of neurons that encode a color-, location-, or motion-specific negative RPE signal either combined across areas (**a**) or split by areas (**b**). Thick blue lines represent averages across both monkeys. Thin continuous gray lines represent data from monkey H, thin dashed gray lines represent data from monkey K. An asterisk indicates $p \leq 0.05$ using a one-sided bootstrap procedure that randomized the feature labels. Dotted lines indicate upper confidence interval. Gray bars indicate chance level proportion at 0.05. **c** Color-tuning indices for each area computed according to Eq. (2). Gray bar represents upper and lower bootstrap confidence interval. An asterisk indicates $p < 0.05$ by falling outside of the specified confidence interval. **d**–**c** equivalent conventions to (**a**–**c**) for feature-specific positive RPE encoding

used an index to quantify the relative proportion of color-selective RPE neurons compared to location- and motion-selective RPEs $[(\mathrm{RPE_{col}} - \mathrm{RPE_{loc+motion}}/2)/(\mathrm{RPE_{col}} + \mathrm{RPE_{loc}} + \mathrm{RPE_{motion}})$, Eq. (2)] with values >0 indicating more prevalent color-specific RPE encoding. This color-tuning index showed that for nRPEs ACC, VS, and dlPFC showed stronger color-tuned RPEs than CD (two-sided bootstrap CI: $p \leq 0.05$; Fig. 6c, see Methods). These results were not dependent on defining nRPE encoding as significantly increased firing with more negative RPE values and hold when nRPE encoding is defined as significantly decreased firing similar to prior studies of midbrain dopamine neurons[30,31]. Overall, ACC and VS are those areas with the largest population of color-specific nRPE information (Supplementary Fig. 7).

Similar to nRPEs, pRPEs were more often color-specific than location- or motion-specific in ACC and VS (one-sided bootstrap CI: $p \leq 0.05$, Fig. 6e, left column). Neurons in CD also selectively encoded feature-specific pRPEs in the color dimension, while neurons in dlPFC were not selective (Fig. 6e, right column). Color-tuning indices did not differ substantially between areas (ACC: $I_{\mathrm{col}} = 0.10$, VS: $I_{\mathrm{col}} = 0.14$, dlPFC: $I_{\mathrm{col}} = 0.05$, CD: $I_{\mathrm{col}} = 0.123$; two-sided bootstrap CI: $p > 0.05$; Fig. 6f).

In contrast to nRPEs and pRPEs, unsigned RPEs were across areas similarly prevalent for the color, location and motion dimensions (one-sided bootstrap CI: $p > 0.05$; Fig. 7a). Split by areas, only the VS encoded surprise signals stronger for color than motion and location (one-sided bootstrap CI: $p \leq 0.05$,

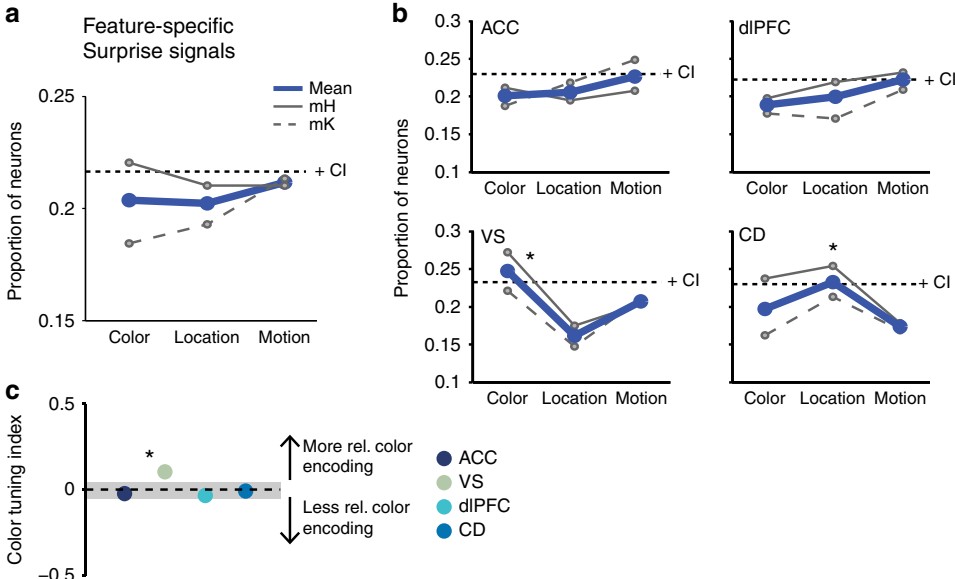

**Fig. 7** Prevalence of feature-specific surprise RPE encoding. Conventions are equivalent to Fig. 6 for feature-specific unsigned RPEs

Fig. 7b bottom left). This finding was confirmed by a significantly higher color-tuning index for VS ($I_{col} = 0.103$) than for ACC ($I_{col} = -0.024$), dlPFC ($I_{col} = -0.036$), and CD ($I_{col} = -0.01$) (Fig. 7c). For a detailed overview of results see Supplementary Table 1 and Supplementary Fig. 8. These results were qualitatively similar with varying statistical criterions for identifying RPE encoding neurons (Supplementary Fig. 9).

**Feature-specific RPEs are segregated from outcome signals.** Feature-specific firing correlations with RPE might emerge from neurons that show already feature-specific firing for outcomes irrespective of prediction error, or they could occur in a segregated neuronal population. To answer this question, we first calculated the prevalence of feature information in the outcome period, by testing whether neurons encoded any of nine feature-related variables using multiple regression analysis (see Supplementary methods). Since neurons likely encoded more than one variable (Supplementary Fig. 3c, d), we evaluated feature information encoding at the first order (greatest regression coefficient) as well as at the second order, in which case a bi-linear regression with two variables had to fit a given neuron significantly better (see Supplementary methods). We found that 20/16/13% of neurons encoded color/motion/location-specific information about the outcome at the first or second order (see Supplementary methods). However, only 35/28/26/% of those neurons also showed significant color/motion/location firing correlations with RPEs, suggesting that prediction error-independent feature tuning during the outcome period of the task does not explain the majority of feature-specific RPE firing (Supplementary Fig. 10).

**Cell-class specificity of RPE encoding neurons.** To understand the mechanisms underlying feature-specific RPEs, it is important to identify the functional cell types encoding them. Our recordings allowed distinguishing two functional cell classes in the cortical brain areas (putative pyramidal cells and putative interneurons), and two cell classes in the striatum (putative medium spiny neurons and putative interneurons) using methods established before[32–34] (Supplementary methods and Fig. 8). The null hypothesis was that the distribution of narrow- and broad-spiking units that encode feature-specific RPEs is the same as the distribution in the total population of recorded neurons. In the

cortical areas ACC and dlPFC, we found that narrow-spiking neurons more likely encoded feature-specific RPE signals (ratio narrow/broad = 0.65) than expected from the total population (ratio narrow/broad in population = 0.41) (chi-square test, $\chi^2 = 5.95$, $p = .015$, $\varphi = -0.096$), while encoding of non-specific RPE signals did not differ (ratio narrow/broad = 0.53; chi-square test, $\chi^2 = 0.37$, $p = 0.55$, $\varphi = -0.027$; Fig. 8c–e). For the striatum we found a statistical trend that feature-specific RPEs were more frequently encoded by narrow-spiking neurons (which include the putative fast-spiking interneurons[34,35]) than suggested based on the population distribution (chi-square test feature-specific RPEs: $\chi^2 = 3.02$, $p = 0.082$, $\varphi = -0.092$) (Fig. 8h–j). Control analyses showed that these results were not merely explained by the higher firing rate of narrow-spiking neurons (Supplementary Fig. 11).

**Feature-specific RPE signaling can affect stimulus selection.** What are the functional consequences of feature-specific encoding of prediction errors? At the behavioral level, prediction errors indicate the need to adjust attention in subsequent trials. At the neural level, this adjustment for future attention might correspond to a shift of firing to the outcome epoch early during learning to firing to the color-onset epoch after learning. Such a temporal transfer of firing from outcome to cue epochs is the classical signature of RPE encoding by ventral tegmental dopaminergic neurons[30,31]. To test whether such a transfer takes place within the population of color-specific RPE encoding neurons, we determined whether the magnitude of the RPE (outcome epoch) in the current trial was related to firing rate changes following color onset in the subsequent trial. We hypothesized that during learning periods when prediction errors are large, neurons would not yet contribute to the visual selection of the color, but after learning, when prediction errors are low, the same neurons would show an enhanced color onset response indicating that they contribute to the attentional selection of the relevant stimulus. We tested this by extracting the 25% of trials with the largest RPE and the 25% of trials with the smallest RPE (trial $n$) for each color-specific RPE encoding neuron, and compared the neurons' change in firing rate from pre to postcolor onset in the trials following those (trial $n + 1$).

We found that color-specific RPE encoding neurons showed on average a significantly enhanced color onset firing (post vs. precolor

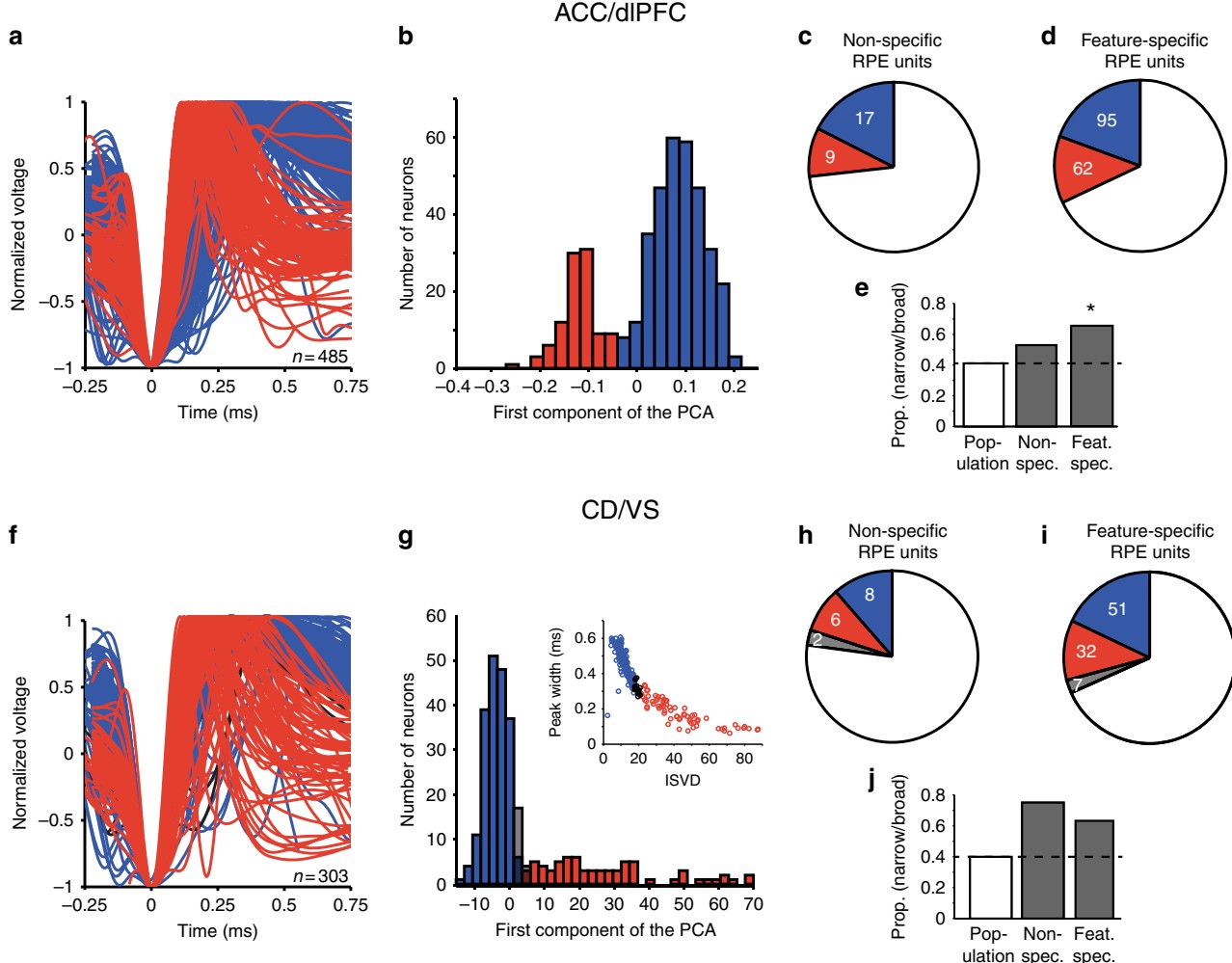

**Fig. 8** Cell-type classification of RPE units. **a–e** for ACC/dlPFC units. **a** Waveforms of all highly isolated single units recorded, identified as putative interneurons (narrow-spiking, red), putative pyramidal cells (broad-spiking, blue). **b** Histogram of the first component of the PCA using peak-to-trough duration and time to repolarization to separate neurons into putative interneurons and putative pyramidal cells. **c** Proportion of nonspecific RPE encoding neurons identified as narrow- (red) or broad-spiking (blue), in addition to nonsingle units (white). **d** Proportion of feature-specific RPE encoding neurons identified as narrow- or broad-spiking. **e** Ratio of narrow to broad-spiking neurons identified in the population, for nonspecific and feature-specific RPE encoding neurons. Black asterisk indicates $p < 0.05$ (chi-square test). **f–j** for CD/VS units. **f** Waveforms of all highly isolated single units recorded, identified as putative interneurons (red) or putative medium spiny neurons (MSNs, blue), or unidentified (black). **g** Histogram of the first component of the PCA using peak width and initial slope of valley decay (ISVD) to separate neurons into putative interneurons and MSNs. Inset shows the scatterplot of peak width versus ISVD across neurons. **h** Proportion of nonspecific RPE encoding neurons identified as putative interneurons or MSNs, or unidentified (gray), in addition to nonsingle units (white). **i** Proportion of feature-specific RPE encoding neurons identified as putative interneurons or MSNs or unidentified. **j** Ratio of putative interneuron/MSN in the population, for nonspecific and feature-specific RPE encoding neurons

onset) (*t* test, $p < 0.0001$ for each RPE type). This increased color onset response was on average stronger following trials with low RPE than high RPE, i.e., after learning (Fig. 2d, Fig. 9c, e, Supplementary Fig. 12), specifically when the preceding trial's choice was for the preferred color of the neuron, i.e., the color for which it selectively encoded a greater RPE signal (Fig. 9c, e cyan vs. gray bars, Fig. 3 top panels). This difference in firing rate change following trials with low vs. high RPE was statistically significant when the preceding trial's choice was for the preferred RPE color, for neurons encoding positive RPE (paired *t* test, $t_{pref} = 3.73$, $p_{pref} < 0.001$, Hedges' $g = 0.28$, $t_{nonpref} = 1.33$, $p_{nonpref} = 0.185$, Hedges' $g = 0.09$, $n = 140$) (Fig. 9c), and for neurons encoding surprise (paired *t* test, $t_{pref} = 3.82$, $p_{pref} < 0.001$, Hedges' $g = 0.19$, $t_{nonpref} = 1.85$, $p_{nonpref} = 0.065$, Hedges' $g = 0.09$, $n = 260$) (Fig. 9e), but not for neurons encoding negative RPE (paired *t* test, $t_{pref} = 1.72$, $p_{pref} = 0.089$, Hedges' $g = 0.13$, $t_{nonpref} = 1.06$, $p_{nonpref} = 0.291$, Hedges' $g = 0.07$; $n = 114$) (Fig. 9a).

The selectively increased firing to color onsets after low RPE trials was most prominent and statistically significant for ACC neurons encoding color-specific positive RPEs (paired *t* test, $t_{pref} = 3.29$, $p_{pref} = 0.002$, Hedges' $g = 0.44$, $n = 44$, Fig. 9d, Supplementary Fig. 12), and for CD neurons encoding color-specific surprise (paired *t* test, $t_{pref} = 344$, $p_{pref} < 0.001$, Hedges' $g = 0.33$, $n = 50$, Fig. 9f, Supplementary Fig. 12). These findings suggest that color-specific RPEs during the early reversal learning trials translate into color cue firing rate increases for these same neurons after reversal learning, reminiscent of the temporal transfer of classical dopaminergic prediction error signals.

## Discussion

We found that about half of the neuronal populations in anterior cingulate cortex, dorsolateral prefrontal cortex, VS, and CD encoded RPEs that were informative about the specific features of the attended and chosen stimulus. This feature-specific RPE was

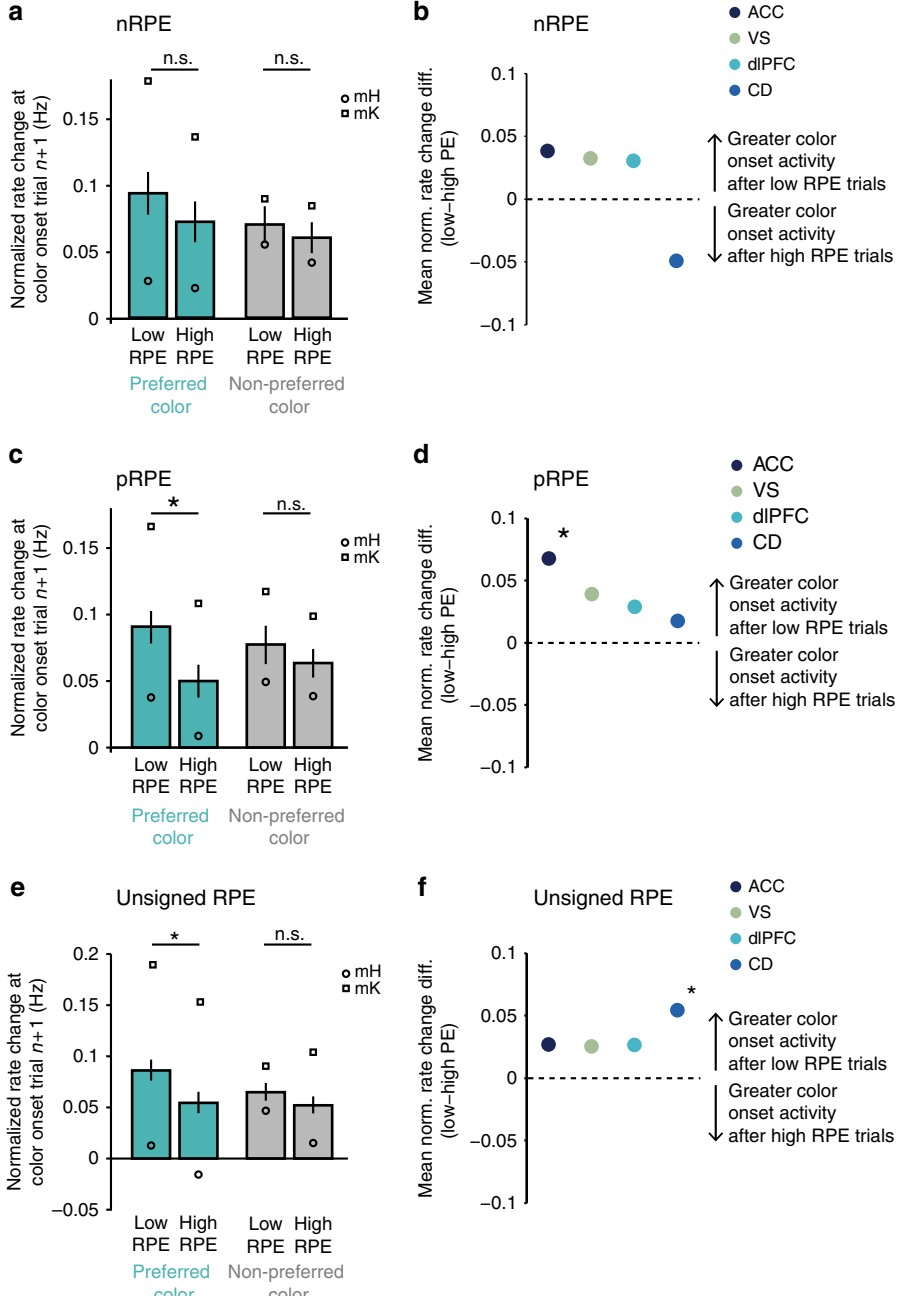

**Fig. 9** Firing rate increases at color onset following low- vs. high-RPE trials. **a, c, e** Average normalized firing rate changes from pre to postcolor onset in trial $n+1$ across neurons encoding color-specific negative RPE (**a**, $n = 114$), positive RPE (**c**, $n = 140$) and surprise (unsigned RPE) (**e**, $n = 260$). Rate changes were computed according to Eq. (13) in Supplementary Methods and normalized to range from −1 to 1. Averages were computed separately for the 25% of trials with the greatest prediction errors and for the 25% of trials with the lowest prediction error, in cyan following preferred color choices and in gray following nonpreferred color choices. Circles indicate means for monkey H, squares indicate means for monkey K. Error bars indicate SEM. **b, d, f** Mean differences in normalized rate changes following low vs. high RPEs for the preferred RPE color (cyan in **a, b**) for each area across color-specific negative RPE (**b**) positive RPE (**d**), and unsigned RPE (**f**) encoding neurons. Black asterisks indicate significant differences in rate changes following low vs. high RPEs (paired *t* test, $p < 0.05$)

more commonly encoded for the task relevant (color) dimension that also predicted reward, illustrating the encoding of the specific, goal-relevant information needed to improve feature-based attention in future trials. Feature-specific encoding of RPEs emerged on average later than nonspecific RPE encoding, indicating that over time it might emerge by combining general prediction error information with feature information.

Among the recorded brain areas, the ACC stood out by containing most neurons with early feature-specific RPE information,

with a slower rise of feature information in the CD, dlPFC, and VS (Fig. 5b). This finding underlines the importance of ACC to provide the specific information needed to adjust attention and behavior in future trials[36,37] and complements previous reports of ACC neurons conveying prediction error related activity for specific actions[38], unique objects[39], stimulus–response mapping rules[40], attentional and motivational origin of errors[36], and more abstract combinations of stimulus and reward information[25]. Our ACC finding uniquely adds to this literature by showing firstly,

that RPE activity in ACC is combined with the attended color feature in a selective attention task that always presented all possible features on the screen. This display induces perceptual ambiguity and makes the task difficult to perform (Fig. 1 and Fig. 6). Secondly, our results show that this feature-specific RPE activity is linked with greater feature-based attention activity in subsequent trials. Neurons that encoded color-specific RPEs increased their activity to the color-cue onset in the next trial (Fig. 9). This finding complements reports of attention-specific activity in ACC[18,41,42] and supports the view that ACC plays a role in controlling to which stimulus features (covert) attention is shifted[43,44].

Our results also clarify that specific information about the source of the RPE, with regards to the currently rewarded feature, is not localized to the ACC, but widely distributed to all four areas we recorded from. These areas are anatomically mono-synaptically connected[2,14,45] and functionally synchronized in different task contexts[41,42,46,47]. The distributed information about feature-specific prediction errors is consistent with the recurrent nature of fronto-striatal processes underlying reward-based choices[48]. In contrast to views that emphasize more localized and serial computations of subprocesses of goal-directed choices[49], acknowledging the recurrent connectivity of fronto-striatal networks entails that many brain areas act in concert to perform similar computations[19,44,48,50]. Consistent with this view, RPEs seem to be evident in many brain regions, in a possibly redundant fashion[51,52] with mixed selectivity for a variety of combinations of task-relevant features[53] (see also Supplementary Fig. 3d). Resonating with such a distributed, mixed code, we found substantial amounts of neurons encoding RPEs for nonreward-relevant stimulus dimensions (motion/location), reminiscent of recent studies in prefrontal cortex showing a prevailing influence of nonrelevant task variables and features over multiple trials[43,54,55]. This encoding of non-relevant information has been suggested to promote behavioral flexibility in volatile task environments by facilitating the detection of unexpected changes in these environments[55,56].

The preponderance of narrow-spiking neurons with feature-specific error information in the cortex and with a statistical trend in the striatum was an unexpected, data driven finding (Fig. 8, Supplementary Fig. 11) that supports suggestions of a particular role of inhibitory neurons to process learning related information and/or to induce plasticity in cortical and striatal networks[57,58]. Narrow action potential waveforms have been associated with fast-spiking inhibitory neurons in cortex and striatum[35,59]. Our finding that putative interneurons are particularly informative about the error term is consistent with their involvement to regulate network level plasticity changes[58], including spike-timing dependent changes at corticostriatal synapses[60] and the balancing of inhibitory with excitatory synaptic strength in balanced networks[61].

For the ACC, dlPFC, CD, and VS, prediction errors correlated with the firing of neurons after correct trials (giving rise to positive RPEs), after incorrect trials (giving rise to negative RPEs), and irrespective of the sign of the actual trial outcomes (giving rise to surprise). Large surprise signals (to rare, high rewards) in the ACC have previously been shown to predict adjustment of behavioral strategies[27], but it has been questioned whether any surprise related activity exists that relates to changes in attention[62]. Here, we refute this view by reporting widely distributed and prevalent neuronal signals conveying surprise for all dimensions of an attended stimulus in ACC, dlPFC, CD, and VS. Most notable was the VS by showing proportionally stronger neuronal surprise signals for the goal-relevant color dimension as opposed to the task-relevant, but reward-irrelevant, location and motion dimensions (Fig. 7c). A feature-specific surprise signal,

that carries no information about the valence of the outcome, reflects the accuracy with which a feature predicts outcome, and therefore indexes the outcome uncertainty associated with a given feature[63]. A long-standing psychological theory of attention suggests that attention during learning is driven by unexpected events such as the surprising outcomes we quantified[6,7]. According to this attention model, unexpected outcomes should give rise to stronger attention to the stimulus feature that gave rise to the violated expectation[64]. Our study tested this hypothesis and found evidence that the same neural population that encodes a color-specific surprise signal also showed stronger firing rate increases after feature onset in later trials of a block when the reward relevance of that color had been learned (Fig. 9). This increased feature selection signal (1) was stronger for the color that was preferred versus non-preferred by the neuron, and (2) it was stronger when subjects had learned the relevant feature, i.e., during trials when prediction errors were comparatively low. These results provide direct evidence for a role of the VS to contribute to attentional biasing towards goal-relevant features, i.e., to "learned attention". This conclusion adds an important functional role to RPE signaling which is—across species—ubiquitously reported to be particularly strong in the VS[65–67]. In contrast, feature-specific surprise signals in ACC, PFC, and CD may primarily serve to increase overall "attentiveness" to all stimulus features during periods of uncertainty to promote correct behavioral adjustment in complex and volatile task environments (Fig. 7).

Our findings also supports recent suggestions that attentional biases reflect an internal activity state within the striatum that resolves competing value predictions and beliefs about possible relevant stimuli[44,68]. In this framework, attention is not considered to reflect a unitary, obscure top-down signal that is localized to the prefrontal cortex as in many classical models, but rather attention emerges from ("is the effect of") the current state of basal ganglia circuits that continuously integrate multiple information streams and resolve competition among these input streams[68]. A core insight from this view is that the striatum has direct access to the spatial maps in the superior colliculus[2] for exerting a direct, ongoing bias for overt sampling and covert attentional selection of visual information (e.g.,[69]). Our study strongly supports this view revealing widespread feature-specific RPEs (Figs. 6, 7) and feature-specific selection effects (Fig. 9 and Supplementary Fig. 12) across the medial (ACC and VS) and the lateral (dlPFC and CD) fronto-striatal loops.

In our task, RPE signals specific for the reward-relevant color are precisely what is needed to enhance those synaptic connections between neurons encoding the specific color that is more relevant than expected, and to reduce the synaptic connection weights among neurons encoding the color that was less rewarded than expected. These types of synaptic weight changes following RPEs have been used in network models implementing various RL rules[22,70]. These models illustrate, for example, that simpler stimulus–response reversal learning performance in monkeys can be realized by spike-timing dependent plasticity changes[71]. However, it has remained unclear how to implement more complex credit assignment in a higher dimensional feature space where multiple features could be credited for an outcome, even though only one feature is actually relevant[4,11]. For this situation, a recent spiking model suggested a four-factor learning rule that is dependent on attention to a specific stimulus feature or action prior to registering a reward/no-reward outcome[22]. In this model, neurons activated by an outcome receive a synaptic tag, which is specific to the attended feature, from feedback connections originating from output neurons similar to striatal output neurons. This attentional feedback-induced synaptic tag acts like an attention-specific eligibility trace that can be combined with

dopamine dependent (feature unspecific) prediction error information when a (rewarding or nonrewarding) outcome is received. Learning is achieved when these two factors (attentional feedback and neuromodulatory prediction error information) meet at the synapses between pairs of neurons that showed near coincident pre and postsynaptic activity during the outcome processing[22]. The models make multiple predictions that our data directly support. Firstly, synaptic updating is taking place in an associative network layer that resembles the fronto-striatal network of value learning as opposed to sensory or motor related network layers. Secondly, feature-specific RPEs should emerge locally in multiple areas across the entire associative network based on neuron-specific synaptic tags, closely corresponding to the distributed RPEs we observed. Finally, the model predicts that learning of task-relevant features depends on deploying attention toward those stimulus features that are most consistently reward associated. This attentional hypothesis of RL was directly tested in our experiment, providing evidence that the most ubiquitously encoded prediction error signals occur for the attended, goal-relevant color feature.

Taking together, our findings support the concept of attention weighted RL as a generic framework to understand learning and attention in environments with multidimensional stimuli[3,4,10]. The existence of network-wide available information about the degree to which individual features of visual stimuli led to unexpected outcomes critically inform learning rules that efficiently solve the credit assignment problem[1,58,67,72]. Our study may thus provide a starting point to understand how network-wide credit assignment processes are directly related to improved biases of feature-based visual attention.

## Methods

**Electrophysiological recordings**. Data were collected from two male rhesus macaques (Macaca mulatta, ages: 7–8 years). All animal care and experimental protocols were approved by the York University Council on Animal Care and were in accordance with the Canadian Council on Animal Care guidelines. Extra-cellular recordings were made with 1–12 tungsten electrodes (impedance 1.2–2.2 MOhm, FHC, Bowdoinham, ME) in anterior cingulate cortex (area 24, ACC), prefrontal cortex (area 46, dlPFC), caudate nucleus (CD), and ventral striatum (VS) through rectangular recording chambers (20 by 25 mm) implanted over the right hemisphere (Fig. 1c). Electrodes were lowered daily through guide tubes using software controlled precision micro-drives (NAN Instruments Ltd., Israel). Data amplification, filtering, and acquisition were done with a multichannel acquisition processor (Neuralynx). Spiking activity was obtained following a 300–8000 Hz passband filter and further amplification and digitization at 40 kHz sampling rate. Sorting and isolation of single unit activity was performed offline with Plexon Offline Sorter, based on principal component analysis of the spike waveforms. Experiments were performed in a custom-made sound attenuating isolation chamber. Monkeys sat in a custom-made primate chair viewing stimuli on a computer monitor (60 Hz refresh rate, distance of 57 cm). Eye positions were monitored using a video-based eye-tracking system (EyeLink, SRS Systems) calibrated prior to each experiment to a nine-point fixation pattern. Eye fixation was controlled within a 1.4°–2.0° radius window. During the experiments, stimulus presentation, monitored eye positions, and reward delivery were controlled via MonkeyLogic (www.brown.edu/Research/monkeylogic/). Liquid reward was delivered by a custom-made, air-compression controlled, and mechanical valve system.

**Behavioral paradigm**. The monkeys performed a feature-based reversal learning task that required covert spatial attention to one of two stimuli dependent on color-reward associations (Fig. 1a). These color-reward associations were reversed in an uncued manner between blocks of trials with constant color-reward association (Fig. 1b). By separating the location of attention from the location of the saccadic response, this task allowed an identification of neural responses to the location of attentional focus independent of neural signals linked to response preparations, during reversal learning. Each trial started with the appearance of a gray central fixation point, which the monkey had to fixate. After 0.5–0.9 s, two black/white gratings appeared to the left and right of the central fixation point. Following another 0.4 s the two stimulus gratings either changed color to green and red (monkey K: cyan and yellow), or they started moving in opposite directions up and down, followed after 0.5–0.9 s by the onset of the second stimulus feature that had not been presented so far, e.g., if after 0.4 s the grating stimuli changed color then after another 0.5–0.9 s they started moving in opposite directions. After 0.4–1 s

either the red and green stimulus dimmed simultaneously for 0.3 s or they dimmed separated by 0.55 s, whereby either the red or green stimulus could dim first. The dimming represented the go-cue to make a saccade to one of two response targets displayed above and below the central fixation point. Please note that the monkeys needed to keep central fixation until this dimming event occurred. A saccadic response following the dimming was only rewarded if it was made to the response target that corresponded to the movement direction of the stimulus with the color that was associated with reward in the current block of trials, e.g., if the red stimulus was the currently rewarded target and was moving upward, a saccade had to be made to the upper response target at the time the red stimulus dimmed. A saccadic response was not rewarded if it was made to the response target that corresponded to the movement direction of the stimulus with the nonreward associated color. Hence, a correct response to a given stimulus must match the motion direction of that stimulus as well as the timing of the dimming of that stimulus. The rationale for this design was to ascertain continuous allocation of attention to one stimulus—since the animal did not know the time of dimming of the current target stimulus (which could occur before, after, or at the same time as the second stimulus), it had to attend continuously until the "Go-signal" (dimming) of that stimulus occurred. If dimming of the target stimulus occurred after dimming of the second/distractor stimulus, the animal had to ignore dimming of the second stimulus and wait for dimming of the target stimulus. A correct response was followed by 0.33 ml of water delivered to the monkey's mouth.

Across trials of a block the color-reward association remained constant for 30 to a maximum of 100 trials. Performance of 90% rewarded trials (calculated as running average over the last 12 trials) automatically induced a block change. The block change was uncued, requiring the subject to use the reward outcome they received to learn when the color-reward association was reversed in order to covertly select the stimulus with the rewarded color. Note that at all times the reward schedule was deterministic. In contrast to color, other stimulus features (motion direction and stimulus location) were only randomly related to reward outcome—they were pseudo-randomly assigned on every trial. Saccadic responses had to be initialized within 0.5 s after dimming onset to be considered a choice (rewarded or nonrewarded). All other saccadic responses, e.g., toward the peripheral stimuli, were considered nonchoice errors.

This task ensured that behavior was guided by the specific color-reward association, which was evident in monkeys choosing the stimulus with the same color following correct trials with 89.5% probability (88.7%/ 90.3% for monkey H/ K), which was significantly different from chance (t test, both $p < 0.0001$). In contrast, monkeys chose the stimulus with the same motion direction following a rewarded trial only with a 46.7% probability (43.7%/46.2% for monkey H/K), and the stimulus with the same location following a rewarded trial with a 44.9% probability (47.3%/46.0% for monkey H/K), indicating a tendency to switch motion and location choices following rewarded trials (t test, all $p < 0.0001$). Although reward was deterministic, performance of the task was not optimal, evident in asymptotic average performance of around 75% (Fig. 2a). This illustrates the general task difficulty, which was likely driven by the need to integrate multiple stimulus features into a single response (Fig. 2, Supplementary Fig. 1).

We used block sine gratings with rounded-off edges for the peripheral stimuli, moving within a circular aperture at 0.8°/s and a spatial frequency of 1.2 (cycles/°) and a radius of 2.0°. Gratings were presented at 5° eccentricity to the left and right of the fixation point.

**Data analysis**. Analysis was performed with custom MATLAB code (Mathworks, Natick, MA), utilizing functions from the open-source Fieldtrip toolbox (http://www.ru.nl/fcdonders/fieldtrip/). All spike-density functions were smoothed with a Gaussian kernel with a standard deviation of 25 ms. Only correct and incorrect choice trials were analyzed, whereby correct choice trials were rewarded trials, while incorrect choice trials were either made to the nonrewarded stimulus or in the incorrect response time window (first vs. second dimming). Fixation breaks, early responses, and no-response trials were not included in any analyses.

Units were only included in any of the following analyses if they (i) had a minimum firing rate of 0.5 Hz within the feedback epoch (0–1.5 s following feedback onset), (ii) prediction errors computed with a RL model (see below) could be computed for ≥40 trials, and (iii) these minimum of 40 trials could be identified as either occurring during learning or after learning according to an ideal observer statistics (see Supplementary methods). All trials from blocks that were not learned to criterion were discarded.

We quantified the trial-by-trial progression of RPEs during reversal performance using a variety of reinforcement learning (RL) principles that were previously found to account for feature-based reversal learning performance[3,4,10,11,23]. We compared models that used prediction errors (RPEs) to update different types of representations using methods similar to a previous study[10], as described in detail in Supplementary methods. The RL models differed how the features were represented and weighted to achieve reversal learning. In a first model, RPEs were used to update all features (e.g., red, green, location left, location right, motion up, and motion down) of a stimulus nonselectively (feature-nonselective RL model, F-NS model), i.e., without using preknowledge about which (feature) dimension is most rewarded. In a second model, RPEs were used to update only the color feature that was systematically linked to reward. This feature-selective RL model (F-S model) assumed that the animals had formed an

attentional set that only included the two different colors as features for a top-down representation. In a third model, we are representing all features as in the F-NS model but include a dimensional weight that learns during a reversal block a higher weight for features of the most reward-consistent dimension (color) and a lower weight for those dimensions that are not systematically linked to reward (location and motion direction). This feature-dimension weighted RL model (F-DW model) thus learns the attentional set during the reversal learning period until the performance asymptotes. We devised a fourth model implicated in previous studies to realize learning with multidimensional stimuli using a decay parameter that reduces the values of features of the nonchosen stimulus. This feature-decay model (F-Dec model) was otherwise identical to the F-NS model. In a fifth model, we combined the decay mechanism with the dimensional weighting mechanism to a feature-dimension weighted decay RL model (F-DW-Dec model) to test whether combining mechanisms of the models improved the fitting of the monkey's learning performance. All models are described in detail in the Supplementary methods.

We optimized the model by minimizing the negative log likelihood over all trials using up to 20 iterations of the simplex optimization method to initialize the subsequent call to fmincon (matlab function), which constructs derivative information. We used an 80%/20% (training dataset/test dataset) cross-validation procedure repeated for $n = 50$ times to quantify how well the model predicted the data. Each of the cross-validations optimized the model parameters on the training dataset. We then quantified the log-likelihood of the independent test dataset given the training datasets optimal parameter values. We found that the F-DW-Dec Model provided the lowest Akaike Information Criterion for both monkeys (Supplementary Fig. 1a), and resulted in the lowest (monkey H) and second lowest (monkey K) (i.e., best) Log-likelihoods for the cross-validated test dataset (Supplementary Fig. 1b, c). These results lead us to choose the F-DW-Dec model for generating prediction errors for the neuronal analysis. The optimized F-DW-Dec Model showed similar parameter values for monkey H/K, with ($\eta$ (learning rate) = 0.22/0.25, $\beta$ (selection noise) = 3.55/2.79, $\phi$ (dimension weighting of feature representation) = 0.68/0.98, and $\omega$ (value decay for nonchosen feature) = 0.92/0.68. These results align well with previous studies using a similar model architecture[3,4,10,11].

To identify RPE encoding neurons, we correlated each neuron's firing rate time-resolved with RPEs obtained from the best fitting RL model (the F-DW-Dec Model). The trial-by-trial development of the average positive and negative RPEs for each monkey are shown in Fig. 2d. We also illustrate the prediction errors for each of the features of the chosen stimulus separately, confirming that the RPE's used by the model to adjust feature values are dominated by the color dimension, while the reward-irrelevant motion and location dimensions show a similar progression of RPE but at a lower magnitude corresponding to their lower feature values (Supplementary Fig. 13). Each correlation analysis required a minimum of 15 trials. We correlated firing rate with positive RPEs in correct choice trials and with negative RPEs in incorrect choice trials. To identify neurons that encoded an unsigned RPE, we used partial correlation analysis to correlate firing rates with the absolute RPE in correct and incorrect choice trials while partializing out the sign of the RPE (by including a co-variate of ±1 for correct/incorrect trials, respectively). The analysis time ranged from −500 to 1500 ms after the outcome event; time windows spanned 200 ms and were shifted by 25 ms. For a neuron to be considered to encode a nonspecific positive, or unsigned RPE signal, it had to significantly positively correlate its firing rate with a positive, or unsigned RPE, respectively (Spearman correlation, $p < 0.05$), for a minimum of four consecutive time bins following the outcome event, while not correlating positively in more than two consecutive time bins before the outcome event. For a neuron to be considered to encode a negative RPE signal, it had to significantly negatively correlate its firing rate with a negative RPE, i.e., the more negative the RPE the higher the firing rate, for a minimum of four consecutive time bins following the outcome event (Spearman correlation, $p < 0.05$), while not correlating negatively in more than two consecutive time bins before the outcome event. A neuron could be identified to encode more than one signal type, e.g., a neuron could encode an nRPE and pRPE. For a supplementary analysis, and to acknowledge previous literature, we also considered the opposite encoding of negative RPEs, with firing rates decreasing with more negative RPEs. In this case, a neuron had to significantly positively correlate its firing rate with a negative RPE for a minimum of four consecutive time bins following the outcome event.

To identify neurons that encoded a feature-specific RPE signal, trials were split into the features of interest prior to the correlation analysis (color, location, and motion direction). The principle for identifying positive, negative, and unsigned feature-specific RPE neurons was the same as for nonspecific RPE signals with additional criteria described in the following. For instance, for a neuron to be considered to encode a color-specific RPE signal, it had to significantly encode an RPE signal (as described above) in minimally four consecutive time bins for trials in which, e.g., color 1 was chosen, while either not encoding or encoding significantly less an RPE signal for trials in which color 2 was chosen. Significant differences between $R$ values (Spearman correlation) for the two trial types were computed by $z$-transforming $R$ values and comparing them using a $z$-test:

$$Z_{\text{observed}} = \frac{z_1 - z_2}{\sqrt{\frac{1}{N_1 - 3} + \frac{1}{N_2 - 3}}}, \tag{1}$$

where $z_1$ and $z_2$ are the $z$-transformed $R$ values for the correlation with feature value 1 and feature value 2, respectively. When $Z_{\text{observed}}$ exceeded $|1.96|$ ($p < 0.05$), $R$ values were considered significantly different for a given time bin. In a minimum of four consecutive bins, $R$ values from correlations with two different feature values (e.g., color 1 chosen or color 2 chosen) had to significantly differ, while an RPE had to be encoded for at least one of the two feature values according to the same criteria as for nonspecific RPE signals. The method of identification was the same for identifying location and motion-specific RPE signals, with the exception of splitting trials according to chosen location or chosen motion direction, respectively. We determined for each neuron the duration in which it encoded an RPE signal as the first span of four or more consecutive significant time bins after the feedback event. Again, a neuron could technically encode more than one signal type, e.g., a neuron could be identified to encode a color-specific pRPE and a location-specific nRPE.

Note that technically a neuron could be identified as encoding a nonspecific and feature-specific RPE. Consider the following example: a neuron may encode a significant RPE for color 1 and color 2 choices, but does so significantly stronger for color 1 choices. This neuron would statistically still "show up" as a nonspecific RPE encoding neuron when color 1 and color 2 choices are collapsed. Since it is not meaningful to label a neuron as feature-specific and nonspecific, for any analysis that explicitly compared feature-specific with nonspecific RPEs (e.g., Fig. 4 and Fig. 8), a neuron was only considered as a nonspecific RPE neuron if it could not also be identified as a feature-specific RPE neuron, making these two separate populations.

To compare time courses of RPE signals, as well as trial outcome signals, we determined for each neuron the time window (minimum 4 consecutive bins) in which it encoded a RPE/trial outcome signal significantly (if a neuron encoded an RPE/trial outcome signal over longer time spans with time bins in between that were not significant, only the first time window of consecutive significant time bins was considered for this analysis). Across neurons, we therefore obtained distributions of time bins in which RPE/trial outcome signals were encoded, and we then tested these distributions for differences in their cumulative sums (Kolmogorov–Smirnoff test, Bonferroni–Holm multiple-comparison correction, $\alpha = 0.05$). To verify these results, we additionally employed nonparametric Rank Sum tests (Bonferroni–Holm multiple-comparison corrected, $\alpha = 0.05$). As an additional measure of latency, we tested whether the time point at which 25% of RPE/trial outcome signals were encoded (the time point when the respective cumulative sum reaches 25%) differed using a randomization procedure ($\alpha = 0.05$, $n = 500$). The analysis procedure was equivalent when comparing the latencies of feature-specific RPE encoding between areas.

We used a bootstrap procedure to determine whether the encoding for any specific feature (e.g., color) was more prevalent than other features (e.g., color versus location and motion direction) based on the distribution across all feature-specific RPEs independent of their specificity (color, location, and motion direction) ($n = 10,000$). Specifically, we assigned each neuron that encoded a feature-specific RPE a value of 1 (color, location OR motion), and every other neuron a value of 0. Across this population, we computed a confidence interval that indicated how likely it was for a neuron to encode a feature-specific RPE (any type). Since we were specifically interested in whether a specific type of feature RPE was encoded more often than the others, we computed a one-sided confidence interval. When the proportion of color-specific RPEs falls above this upper confidence interval, it indicates that color-specific RPEs were more often encoded than would be expected based on the population of all feature-specific RPEs (Figs. 6, 7). This bootstrap procedure was computed across all units encoding a specifically signed or unsigned RPE, initially independent of area recorded (Figs. 5a, d, 6a), and in a second step separately for each area (Figs. 6b, e and 7b). To compare the ratio of color-specific RPE encoding versus location- or motion-specific RPE encoding between areas, we computed a color-tuning index for each area as follows:

$$I_{\text{col}} = \frac{P_{\text{col}} - (P_{\text{loc}} + P_{\text{mot}})/2}{P_{\text{col}} + P_{\text{loc}} + P_{\text{mot}}}, \tag{2}$$

whereby $I_{\text{col}}$ refers to the color-tuning index, $P_{\text{col}}$, $P_{\text{loc}}$, and $P_{\text{mot}}$ refer to the proportions of color-, location-, and motion-specific RPE units, respectively. We then compared color-tuning indices across areas by computing a two-sided confidence interval (bootstrap procedure, $n = 10,000$) around color-tuning indices that were computed with randomized area labels. An area was considered to have a significantly greater or smaller color-tuning index than the other areas if it fell outside of the confidence interval (Figs. 6c, f, 7c).

**Reporting summary**. Further information on experimental design is available in the Nature Research Reporting Summary linked to this article.

## Data availability
All data supporting this study and its findings, as well as custom MATLAB code generated for analyses, are available from the corresponding author upon reasonable request.

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

## Acknowledgments

This work was supported by grant MOP 102482 from the Canadian Institutes of Health Research (T.W.) and the Natural Sciences and Engineering Research Council of Canada (T.W.), as well as by the Brain in Action CREATE-IRTG program (M.O. and T.W.), and by grant LPDS 2012-08 from the Deutsche Akademie der Naturforscher Leopoldina (S. W.). Imaging data provided by the Duke Center for In Vivo Microscopy, an NIH Biomedical Technology Resource (NIHP41EB015897, 1S10OD010683-01). The funders had no role in study design, data collection and analysis, the decision to publish, or the preparation of this manuscript. The authors would like to thank Hongying Wang for technical support.

## Author contributions

T.W. conceived the experiment. T.W. and S.W. designed the experiment. M.O., S.W., M.A., and S.A.H, performed the experiments. M.O., T.W., S.A., and P.T. analyzed the data. M.O and T.W. wrote the original draft. All authors edited the paper.

## Additional information

**Competing interests:** The authors declare no competing interests.

