## [Peer Review File · Nature Communications]

Reviewers' comments:

Reviewer #1 (Remarks to the Author):

The manuscript presents a rich behavioural experiments combined with electrophysiological recording from four fronto-striatal brain regions in primates. The behavioural task includes visual stimuli with different colours, motion direction and position wherein the reward is contingent on the stimulus colour and it reverses in blocks of trials. Electrophysiological recordings have revealed neuronal responses, in particular at the time of outcome, which could resemble prediction errors driven from a Bayesian-RL model.

One major issue with the behavioural paradigm, in relation to specific aims and analyses of the paper is that there is only one stimulus feature (stimulus colour) which is predictive of reward. An experiment which aims to reveals feature-specific prediction errors would ideally need to make rewards conditional to different features of the stimulus in different blocks, which is not the case here. As such, I am not yet convinced how the present behavioural design could be used to reach conclusions about neuronal encoding of feature-specific prediction errors.

On the other hand, the manuscript has produced an elegant paradigm in which animals are performing a reward reversal task but the action suitable for reporting the choice is signalled in a trial-by-trial fashion. That is, for a red (rewarded) stimulus appearing on the left side of the monitor, the monkey should make an upward or downward saccade depending on the stimulus motion direction in that trial. This is producing a nice situation in which reward prediction is decoupled from action preparation, an important feature which has been absent in many previous studies. Given this, I am wondering why the manuscript is not taking advantage of this aspect of the behavioural design and focus on neuronal encoding of predicted reward and prediction errors independent of the underlying action selection and preparation.

Specific comments:

The manuscript introduces a RL model used for estimating prediction errors but only presents one schematic and abstract figure of this model. The underlying variables of this model need to be presented more extensively, and most importantly, more intuitively in the main text. It would be important to make a plot of few consecutive trials and show the quantity of the prediction errors. What do the feature-specific and non-specific prediction errors look like in these trials?

The RL model has been cross-validated against a model without Bayesian part and a model that does not update the value of the unchosen option. That is fine but if the point is to think about feature-specific prediction errors, then the model should be cross-validated against other models, in particular a model that only computes prediction errors based on colours.

Related to behaviour/model, the manuscript should examine whether the irrelevant features have any influence on the choices of monkeys. For instance, if a particular motion direction was rewarded in the previous trial, is it the case that monkeys make more frequent choices towards the same motion direction in the next trial? If so can the model account for this? Extensive analyses of this type could reveal the relative effects of stimulus features on choices and could be tested against the neuronal data.

The main text will require more information about neuronal data analysis, so that the reader can get a general idea about the analysis without reading through the methods section.

Neuronal analyses includes an arbitrary threshold (of 4 consecutive time bins) to identify neurons with significant encoding. I wonder about the robustness of this form of analysis. How much of the results would hold if a more conservative threshold of say 6 was chosen? Looking at example neurons presented in Figure 2, this becomes particularly concerning. For instance, for neurons presented in V the time windows which are marked as significant (grey bar) seem random. The first neuron in V show non-overlapping error bars prior to the grey window and for the lower neuron in V the responses are completely overlapping in the time interval shown by the grey bar. Lower neurons in I and II have similar issues. This is a major issue since an analysis which identifies these as significant differences seem arbitrary and not conservative enough.

For identifying neurons that encode negative prediction error, the manuscript sets up the analysis so that the more negative the prediction error, the higher neuronal firing rate should be. Given previous works on prediction error coding, it would be necessary to also assume the opposite situation, that is the higher the negative prediction error, the lower the neuronal activity.

Minor

: The last paragraph of the introduction is making confusing statements about prediction error and eligibility trace

Lines 303-305: the sentence is unclear

Figure 8a-b : too many lines and arrows does not allow the reader to easily understand the figure

Reviewer #2 (Remarks to the Author):

Oemisch M et al., Feature-specific prediction errors and surprise across macaque fronto-striatal circuits during attention and learning

The authors describe a study in which they trained monkeys to learn that one of two colored stimuli was relevant within a block of trials. The animals had to indicate whether a drifting grating within the relevant stimulus was moving up or down, and then make a saccade in the corresponding direction. The color of the relevant stimulus changed at intervals, and the location of the stimulus was irrelevant throughout the experiments. While the animals carried out the task, single neuron activity was recorded from dlPFC, ACC, the caudate and ventral striatum. They found prediction error signals that were modulated by irrelevant feature dimensions, which they referred to as stimulus specific.

This paper addresses an interesting question, about the effects of task-irrelevant stimulus dimensions on prediction error signals across multiple frontal-striatal circuits. The authors have recorded from an impressively large number of neurons, targeting important neural circuitry for this task. The results are clearly presented. I have a number of comments, however, which could help clarify the manuscript.

Comments

1. I was somewhat surprised that the authors did not reference the literature on intra- vs. extradimensional set shifting. This has been extensively studied in prefrontal cortex, primarily by Trevor Robbins group in Cambridge. It seemed relevant to the introduction as it was developed.
2. That being said, I found the introduction to be a bit out of step with the actual task on which the animals were trained. While the intro frames the paper as being relevant to understanding how the animals learn which features of a stimulus are important, the task only varies the color which should be attended to, to make a choice. Therefore, the animals are not required to learn or switch among different dimensions within the experiment. Clearly they had to learn this during training, but they are not required to do it during the recordings. And, the animals are over-trained, so at this point

they are likely only attending to color. It took me a while to work out what the task was, mostly because I had formed a different expectation after reading the introduction.

3. Several studies have shown that prefrontal cortex (mostly dlPFC) contains a representation of task irrelevant stimulus dimensions (for example Constantinidis and Goldman-Rakic, as well as Mante, Sussillo and Newsome). This is also likely related to mixed selectivity, although it is possible to have one without the other.

4. It should be made clear that rewards were deterministic, when the task is described at the beginning of the results. While this is a learning task, the paradigm is more consistent with the cognitive literature than the reinforcement learning literature.

5. The reward outcome is largely confounded with the choice in this task. I guess they can be dissociated after reversals. Is there a clear temporal signature of this?

6. I was not clear about why positive and negative reward prediction errors were separated. Why not just assess whether neurons encoded either signed or unsigned RPEs? Also, perhaps the neurons were encoding chosen value and reward, instead of RPE, since RPE can be calculated from these two quantities. This should be assessed.

7. The results in Fig. 3 were unconvincing. Such a small number of neurons encoded non-specific RPEs that it seems to be an irrelevant feature of the neural response. Comparison of onset timing of this feature with timing of feature specific RPEs does not seem meaningful.

8. What p-value was used to assess significance for Fig. 3?

9. I would refer to PFC as dlPFC, since prefrontal cortex is usually defined broadly as OFC, ACC, dlPFC, vlPFC, etc.

10. It seems there may be a firing rate confound in the cell-type analysis. High firing rate neurons encode more information than low firing rate neurons. What if firing rate is controlled for in the analysis which compares coding in excitatory and inhibitory neurons?

Reviewer #3 (Remarks to the Author):

Oemisch et al. report in this manuscript findings from a study of feature-specific reward prediction error (RPE) signals in four different brain areas (two cortical: ACC & LPFC, two striatal: VS & CD). The topic of the study is important and timely, and the experimental effort is exceptional (1960 units recorded in 4 different brain areas in two monkeys). However, I have a few comments and questions that should be addressed.

Main concerns:

1.) I do not fully understand the two hypothesis that are tested in this paper. Describing them more clearly and contrasting the findings in this study with the previous one in PFC would be very helpful to understand the larger significance of the paper.

2.) I do not completely understand the task. Specifically, the description of the task in the methods section does not completely fit what is shown in figure 1a. For example, why do the gratings change color to black/green and black/red (for one monkey, different colors for the other monkey)? Why not red/green? If one of the gratings is black, it is obvious, which grating should be followed. That reduces the effort of allocating attention correctly. More importantly, I do not understand the section coming after that (starting on page 21) that describes a complex sequence of changes in the displays. I do not understand what is going on and in which temporal order. A comprehensive figure might help. More importantly, I do not understand the rationale for this task design. Lastly, I do not understand why the authors use the complex dimming pattern in their task. Again, what is the rationale?

3.) Since the reversals were only in the color domain, the monkeys should have learned to completely ignore the other stimulus features (location, motion). Was that the case? The authors never describe the final best-fitting model and what the relative weight of the different feature dimensions is. This would be interesting in two ways. First, if the monkeys still responded to RPE in the non-relevant dimensions, one has to ask why the monkey was unable to learn to ignore them. Second, if the monkey learned to ignore them, the question arises, why there were feature-specific RPE signals for non-relevant dimensions.

4.) The authors should include a table with the exact number neurons that carry RPE signals for a specific feature.

5.) More importantly, it is not clear to me how specific a feature-specific RPE signal really is. Do these neurons only carry RPE signals for one feature dimension (and no other)? Within a feature dimension, are the neurons specific for only one out of two possible types (as implied in Fig. 2) or is there more variety. Asked another way, are the feature specific and unspecific RPE signals two very different, non-overlapping classes or are they along a spectrum, where some cells are very specific, some cells are specific for a few features and some cells for all features? The authors need to better document their results in that regard.

6.) The meaning of 'negative' and 'positive' RPE is slightly unclear. Is it a neg/pos correlation of firing rate with RPE along its entire axis, or neurons that encode only negative or positive deviations from expectations (but not deviations with the other sign)?

7.) The idea that feature-specific RPE signals follow unspecific ones is not supported by the data. In the earliest time bins there are already a substantial number of feature-specific neurons. The authors need to be more careful with their conclusions.

8.) Lastly, the firing pattern dynamic that the authors interpret as evidence for reinforcement learning in the attention domain is so unspecific that most learning processes would produce it. This hypothesis is not convincingly supported and should be taken out of the paper.

Point-by-point Reply - Summary of Changes in the Revised Text:

We addressed the reviewers' comments in detailed point-by-point replies by adding new analyses, adding new figures and methods sections, and by clarifying the text.

We summarize here the major changes in the revised manuscript in response to the reviewer comments. The revisions did not change any conclusions from our original manuscript. We believe these changes enhanced the manuscript and are grateful for the reviewers' insights.

1- Concerns about the learning task: We outline in our reply to the reviewers why we believe the strength of our highly controlled feature-based learning task might not have been well described by us in our original submission.

We clarify in the revised manuscript and in the replies to the reviewers that our task allows us to report two major novel results: First, it allowed finding that feature-specific prediction errors are neuronally encoded for all feature dimensions that are task-relevant and attended (the location and motion direction and color of the stimulus). Secondly, it allowed identifying that the feature dimension that determines which stimulus will be attended in future trials (i.e. color) gives rise to significantly more feature-specific prediction error encoding neurons than the other dimensions. The manuscript delineates how these findings are encoded at different times in four brain areas important for learning.

These are novel and important findings, because they clarify that the prediction error encoding carries sufficiently specific information to bias future attention. Our finding is particularly important for researchers combining the larger field of attention with the field of learning and memory - a field with a growing importance as exemplified by e.g. Roelfsema, P.R., and Holtmaat, A. (2018). Control of synaptic plasticity in deep cortical networks. *Nature Reviews Neuroscience* 19, 166-180.

2- Terminology: We clarified how features and feature dimensions are defined and adjusted the terminology throughout the manuscript. The term feature is used as in studies about feature-based attention (see e.g. Treue, S., and Martinez Trujillo, J.C. (1999). *Feature-based attention influences motion processing gain in macaque visual cortex. Nature* 399, 575-579). This terminology is critical for us as we believe that our results critically contribute to understanding how feature-based biases of attention are established in neural circuits.

3- Example Neurons: We added more examples (new Fig. 2e) showing feature-specific prediction error encoding and show in Fig. 3 examples for prediction errors specific to each feature dimension, i.e. we show neurons encoding RPEs that are color-specific, location-specific, and motion-specific.

4- Model Details: We added a more explicit description of the models to the main results text and provide a new extensive cross validation analysis of five learning models using log-likelihood optimization (and Akaike's Information criterion to compare models with different numbers of parameters). These new results strongly support that the two monkeys: a) tracked individual features (i.e., that a specific color was systematically rewarded in a series of trials, whereas e.g. upwards motion or left location were not), and b) weighted information about feature dimensions (i.e., color is prioritized against motion and location).

5- Validation of Latency Analysis: We added analyses to add further validation to our finding that feature-specific prediction errors are encoded on average later than non-specific prediction errors (and later than reward outcome per se). Specifically, we show that our statistics are robust against the specific number of consecutive time bins considered for significant neurons.

We explain in detail that in total, we provide now three independent analyses supporting and validating our main finding (see our text for details and replies to reviewers).

6- Cell-type specificity: We added three control analyses showing that higher average firing rate does not explain why narrow spiking neurons (putative interneurons) carry more feature-specific prediction error information than previously thought (new Figure S9). More generally, our data-driven approach highlights the role of narrow spiking interneurons in prediction error processing, providing a critical link between cell-specific insights and network level modeling of learning mechanisms, which is unprecedented in the literature.

7- Validating the Reward Prediction Error Definition: We added a further analysis (and Fig. S5) to show how our finding (ACC and ventral striatum encoding most prominently color specific negative prediction errors) also holds when using an alternative definition of "negative prediction error" that was previously applied to midbrain neurons.

8- Adding Details. At multiple places in the text we enhanced the reading by providing more explicit information about methods used (e.g. the time window durations of analyses and the number of neurons considered in statistical tests). We also added a figure (S6) and a Supplementary Table 1 to show details about the total number of neurons and overlap of neuron populations encoding the prediction error and reward outcomes.

Reviewer #1 (Remarks to the Author):

Comment 1:

The manuscript presents a rich behavioural experiments combined with electrophysiological recording from four fronto-striatal brain regions in primates. The behavioural task includes visual stimuli with different colours, motion direction and position wherein the reward is contingent on the stimulus colour and it reverses in blocks of trials. Electrophysiological recordings have revealed neuronal responses, in particular at the time of outcome, which could resemble prediction errors driven from a Bayesian-RL model.

One major issue with the behavioural paradigm, in relation to specific aims and analyses of the paper is that there is only one stimulus feature (stimulus colour) which is predictive of reward. An experiment which aims to reveal feature-specific prediction errors would ideally need to make rewards conditional to different features of the stimulus in different blocks, which is not the case here. As such, I am not yet convinced how the present behavioural design could be used to reach conclusions about neuronal encoding of feature-specific prediction errors.

Response: We would like to answer in four steps and hope to convey that our findings are showing "feature-specific" effects in a sophisticated task that disentangles the influence of different features. The study of "dimension-specific" prediction error modulation is subject of ongoing other studies but was not driving our study here.

- 1) *We denote a neuron to encode feature-specific prediction errors when the neuron changed the firing rate for one feature (e.g. upwards motion) but showed no rate modulations for the other feature (e.g. downwards motion) of the same feature dimension. The paper describes that these feature-specific modulations occur for each of the three feature dimensions (color, motion, location) to a significant extent in all*

areas. This happens regardless of whether a feature-dimension is systematically linked to reward.

A parsimonious explanation why this occurs is that all features of the chosen stimulus are attended: The color should determine which stimulus is attended. The motion direction has to be attended to program the correct saccade. The stimulus location is attended to focus on the correct peripheral stimulus and the corresponding go-signal.

In addition to these effects we find that after task training the features that were systematically associated with reward (color red or green) lead to significantly more neurons showing feature-specific prediction error modulations. Particularly in ACC, VS and caudate, but also in dlPFC for positive prediction errors.

We believe that these findings do show feature-specific prediction error modulation. We validate these findings in multiple statistical approaches and hope the reviewer would appreciate that our conclusion to report feature specific prediction error encoding is valid.

- 2) We apologize if our original terminology was confusing and we clarify in the revised manuscript at multiple locations (marked with red font) how we use the term “feature-specific”.

We follow in our manuscript the tradition of the feature-based attention literature with our terminology, which frequently isolates feature-specific attention effects when e.g. attention is directed to up- versus downward motion direction irrespective of other stimulus feature dimensions, see e.g. Treue, S., and Martinez Trujillo, J.C. (1999). Feature-based attention influences motion processing gain in macaque visual cortex. *Nature* 399, 575-579.

In other words, features have a dimension (color, direction of motion) and a value (red, upwards). We abbreviate “feature value” by feature to align our paper with the feature-based attention literature, to not clutter the text, and to prevent confusion with the reinforcement learning literature (where “value” has its own connotation).

- 3) The reviewer wonders whether we should not have other feature dimensions than color reward-associated to reach interesting conclusions. We believe points 1 and 2 suggest that we can reach interesting and novel feature-specific prediction error conclusions with our design. However, to switch reward to different feature dimensions will be interesting and important in future studies: Our task was particularly demanding for the animals to learn and perform and this specific paradigm did not allow macaque monkeys to switch the reward association from one feature-dimension to another dimension without compromising other components of the task. In future designs, we will achieve this dimension shifting, which we believe will result in “dimension-specific” prediction error modulation. So far, we found that one way to simplify the task to allow monkeys to learn to switch between feature dimensions might be to use a more natural foraging task and more naturalistic stimuli (rather than gratings) as we propose in Watson, Voloh, Naghizadeh, Womelsdorf, (2018). Quaddles: A multidimensional 3D object set with parametrically-controlled and customizable features. *Behavior Research Methods*.
- 4) The RL model we found to be the best fitting model suggests that both, feature dimension weighting (of color, location, and motion), as well as specific feature values (red color, upwards motion) are used by the monkey to learn the reversals. In the revised

manuscript, we added extensive model comparisons (new Fig. 2, Fig. S1, and Suppl. Methods) to reveal this. Dimensions are used to weight the updated values in the model. Features are used to decay the values of the nonchosen stimulus.

We believe that these model results are an additional argument why our findings of feature-specific firing rate modulation are novel, valid, and important.

Comment 2:

On the other hand, the manuscript has produced an elegant paradigm in which animals are performing a reward reversal task but the action suitable for reporting the choice is signalled in a trial-by-trial fashion. That is, for a red (rewarded) stimulus appearing on the left side of the monitor, the monkey should make an upward or downward saccade depending on the stimulus motion direction in that trial. This is producing a nice situation in which reward prediction is decoupled from action preparation, an important feature which has been absent in many previous studies. Given this, I am wondering why the manuscript is not taking advantage of this aspect of the behavioural design and focus on neuronal encoding of predicted reward and prediction errors independent of the underlying action selection and preparation.

Response: We thank the reviewer for the appreciation of our task design. Our statistical analyses take into account that color-specific effects are disentangled from motion- and location-specific effects. We believe that we utilize the strength of the paradigm already in the current manuscript. We would like to convey that various additional analyses of predicted reward (including prior and during the choice) will be interesting venues for future studies. We do not see how analyses in that direction would not increase the scope of our current manuscript too much. Our manuscript's aim and rationale is to document feature – specific prediction error correlations in the four recorded brain areas. We believe this is an extensive set of statistics with results that will be very important to constrain computational learning models and feature-based attention mechanisms.

Comment 3: The manuscript introduces a RL model used for estimating prediction errors but only presents one schematic and abstract figure of this model. The underlying variables of this model need to be presented more extensively, and most importantly, more intuitively in the main text. It would be important to make a plot of few consecutive trials and show the quantity of the prediction errors. What do the feature-specific and non-specific prediction errors look like in these trials?

Response: We agree with the reviewer's point of view and are grateful for the opportunity to add more model information. We

- (1) added more detail about the modelling in the main text, including which models we compared to find the model best accounting for the learning behavior,*
- (2) added trial-by-trial examples and average distributions of prediction errors (new Fig. 2),*
- (3) added neuron examples about how feature specific prediction error encoding looks like (Fig. 2, but also see the examples already in Figure 3), and*
- (4) added details about model evaluations for both monkeys (Suppl. Methods section and Fig. S1).*

In the main text we introduce the models by writing:

“In order to discern trial-by-trial neuronal encoding of reward prediction errors (RPEs) we evaluated various reinforcement learning (RL) models using RPEs to account for feature-based learning performance^{5,13} (**Suppl. Methods**). The models used RPEs to update values either (1) for all stimulus features independently, (2) for only the different colors that were systematically linked to reward, or (3) for all features but weighted by a dimensional weight. In addition, we evaluated a Bayesian learning model that tracked reward expectations for each stimulus feature without the use of RPEs. We found that the monkey’s learning performance was best explained by the model that learns to weight values of specific features by their dimension, i.e. that learned a location, motion direction and color weight, and additionally decayed feature-specific values of the non-chosen stimulus. These findings corroborate previous studies^{5,13} suggesting that both dimension and feature information contribute to learning performance (**Figure 2b, Figure S1, Suppl. Methods**). The Dimensional-weighting of features plus decay model was superior to models that either treated all features non-selectively, or that only included the color dimension by artificially removing the influence of the location and motion dimension. This was evident for both monkeys in lowest Akaike Information Criteria scores across models and low cross-validation log-likelihood values (**Figure S1**). We thus used that model to generate trial-by-trial estimates of RPEs as shown in **Figure 2c**, allowing us to correlate neuronal firing rates during the -0.5-1.5 sec. reward outcome epoch of each trial time-resolved in 200ms windows shifted by 50ms (**Figure 2d, e**).“ (p. 7)

Comment 4: The RL model has been cross-validated against a model without Bayesian part and a model that does not update the value of the unchosen option. That is fine but if the point is to think about feature-specific prediction errors, then the model should be cross-validated against other models, in particular a model that only computes prediction errors based on colours.

Response: We agree. To address the reviewer’s suggestion, we have added details about the cross-validation and model evaluation procedures for the model used in the original paper and for other models we added in response to the reviewer’s comment, including:

- 1) a model only using color as proposed by the reviewer. We have in prior publications and a similar task shown that a “color only” model is a good but not the best fitting model (Balcarras, M., Ardid, S., Kaping, D., Everling, S., and Womelsdorf, T. (2016). Attentional Selection Can Be Predicted by Reinforcement Learning of Task-relevant Stimulus Features Weighted by Value-independent Stickiness. *J Cogn Neurosci* 28, 333-349.).
- 2) a model that uses all features equally, i.e. independently;
- 3) a model that weights the dimensions of features after updating, and
- 4) a model that weights the dimensions of features after updating, but prior to that dimension weighting of values applies a decay of the values of the non-chosen feature. This decay component is feature-specific and improves the model fit despite adding a free parameter as revealed by lower Akaike Information criterion (new Fig. S1).
- 5) We also added as baseline a Bayesian learning model that solves the task without using prediction errors in order to quantify that such a model is not a good fit. This result seems important (and was missing) to illustrate that prediction errors are very likely used to solve the reversal task.

The results show and quantify that the model we used originally to generate prediction errors is the model that performs best considering the Akaike Information criterion in both monkeys (Fig.

S1a). It uses a dimension weight for the feature values, and it uses a decay of values for individual features.

Comment 5: Related to behaviour/model, the manuscript should examine whether the irrelevant features have any influence on the choices of monkeys. For instance, if a particular motion direction was rewarded in the previous trial, is it the case that monkeys make more frequent choices towards the same motion direction in the next trial? If so can the model account for this? Extensive analyses of this type could reveal the relative effects of stimulus features on choices and could be tested against the neuronal data.

Response: We truly appreciate the reviewer's suggestion, but do not believe that the choice history analysis is needed to arrive at valid conclusions about prediction error encoding in our task. We have however addressed the possible confound of "sticky" choices as follows.

Our task design took care to randomize the motion direction, location and color to the stimulus from trial to trial. As such motion and location had on average a random correlation with rewarded choices, and hence on average repeated choices of the stimulus with the same motion direction or the same location were close to random.

To show this explicitly in the manuscript and address the reviewer's suggestion, we added this analysis result to the manuscript. In particular, we found that following a correct trial choices to the same versus to a different motion and location as in the previous trial were on average 45% and 47%, respectively (close to the chance level at 50%). For color this proportion was 90%, indicating that the reward association of a specific color guided the behavior, while animals were slightly more likely to switch the location and motion direction following a correct trial. We added these findings to the revised manuscript:

"Following a rewarded trial, monkeys chose the stimulus with the same color in the next trial with an 89.5% probability (88.7%/ 90.3% for monkey H/K), which was significantly different from chance (t-test, both $p < .0001$), indicating that the reward association of a specific color guided behavior. Monkeys chose the stimulus with the same motion direction following a rewarded trial only with a 46.7% probability (43.7%/ 46.2% for monkey H/K), and the stimulus with the same location following a rewarded trial with a 44.9% probability (47.3%/ 46.0% for monkey H/K), indicating a tendency to switch motion and location choices following rewarded trials (t-test, all $p < .0001$)." (p. 6)

We believe that the proposed additional modelling is outside of the scope of the current manuscript and is better applied to a probabilistic learning task in which choice history and reward history can statistically be cleanly disentangled (our task used a deterministic reward schedule, which we describe more prominently in the revised text). The focus of this manuscript is on identifying neural correlates of prediction error signals that explain future choices, rather than analyzing the sources of interference on current selection. A follow-up analysis for a separate manuscript (work in progress) specifically focuses on disentangling these aspects.

Comment 6: The main text will require more information about neuronal data analysis, so that the reader can get a general idea about the analysis without reading through the methods section.

Response: We added at multiple locations in the results section more details about the analyses to convey information e.g. on the time windows used, the definition of conditions, the rationale

for a statistical analysis etc. We thank the reviewer for suggesting this. All changes in the revised documents are in red font.

Comment 7: Neuronal analyses includes an arbitrary threshold (of 4 consecutive time bins) to identify neurons with significant encoding. I wonder about the robustness of this form of analysis. How much of the results would hold if a more conservative threshold of say 6 was chosen? Looking at example neurons presented in Figure 2, this becomes particularly concerning. For instance, for neurons presented in V the time windows which are marked as significant (grey bar) seem random. The first neuron in V show non-overlapping error bars prior to the grey window and for the lower neuron in V the responses are completely overlapping in the time interval shown by the grey bar. Lower neurons in I and II have similar issues. This is a major issue since an analysis which identifies these as significant differences seem arbitrary and not conservative enough.

Response: Firstly, we show in the revised manuscript that with a more conservative threshold of 6 consecutive time windows we found that results were remarkably consistent. We added these results on p.8-10 and show them in newly added Suppl. Figures S4 and S7.

Secondly, we need to clarify a misunderstanding of Figure 3 (originally Figure 2) that has led to some confusion for which we apologize. Each column in Figure 3 displays the data from one neuron, meaning there are a total of 6 neurons displayed in Figure 3. For instance, neuron 1 (first column, VS neuron) displays a clear preference to fire more when a choice ended in a large RPE compared with a small RPE (first row, firing rate differences between very light and dark blue condition) when that choice was color 1. When the animal chose color 2 (second row) the neuron did not differentiate between RPE sizes (overlapping spike density functions). The difference between RPE encoding for color 1 choices versus color 2 choices is illustrated in row 3, which displays the actual statistics – the correlation between firing rate and RPE for color 1 choices (black line) and the correlation between firing rate and RPE for color 2 choices (grey line). This is where the grey bar comes from – the statistical difference between the R-values that result from the correlation between FR and RPE for color 1 choices versus the correlation between FR and RPE for color 2 choices.

What is then quite visible for this example neuron is that the statistical measure is already quite conservative – in as few as 4 consecutive time windows were the FR-RPE correlation significantly different between color 1 and color 2 choices (grey bar + red stars).

We clarified the explanation of Figure 3 by adjusting the text (for the first neuron) in the following:

*“For instance, the VS neuron in **Figure 3I** responded from weak to strong when the absolute RPEs (surprise) were low to high when color 1 was the chosen feature value (upper panel), but showed no firing modulation with RPEs when color 2 was the chosen feature value (middle panel), resulting in a significant color selective RPE - firing rate correlation in the 100-200ms following reward onset (bottom panel). Similar examples were evident for other feature dimensions. The ACC neuron in **Figure 3IV** showed stronger firing with more negative RPE, but only when the selected stimulus was located on the left (top panel), and not when it was located on the right (middle panel). Finally, the dIPFC neuron in **Figure 3V** fired stronger with larger pRPEs when the motion of the chosen stimulus was upwards (top panel), but not when it was downwards (middle panel).” (p. 8-9)*

Comment 8: For identifying neurons that encode negative prediction error, the manuscript sets up the analysis so that the more negative the prediction error, the higher neuronal firing rate should be. Given previous works on prediction error coding, it would be necessary to also assume the opposite situation, that is the higher the negative prediction error, the lower the neuronal activity.

Response: We agree with the comment and addressed them by adding an analysis in which we defined negative RPE encoding neurons as those that showed a decrease in firing rate with increasing (more negative) RPE values. Results were similar and are displayed in a newly added Figure S5 with the following main text in the revised manuscript:

“To ensure that these results are not dependent on our definition of negative RPE encoding (defined as significantly increased firing with more negative RPE values), we repeated the analysis for those neurons that may encode a negative RPE with decreased firing similar to midbrain dopamine neurons^{40,41}. Results were comparable and showed that ACC and VS are those areas with the largest population of neurons that encode negative RPEs in a color selective manner (Figure S5).” (p. 12)

Comment 9: Minor

: The last paragraph of the introduction is making confusing statements about prediction error and eligibility trace

Response: We apologize for the confusion and adjusted the paragraph to read the following:

“We found that substantial proportions of neurons in all brain areas tested encoded a prediction error that was informative about the specific feature that was just chosen (e.g. red) and the strongest encoding was evident for the task- and reward-relevant stimulus dimension (color). These feature-specific prediction errors may therefore guide synaptic learning across the entire fronto-striatal loop. We found that these feature-specific prediction errors emerged on average after an initial unspecific prediction error and were associated with stronger attentional selection signals in subsequent trials, thus potentially contributing to improved learning and visual selection.” (p. 4-5)

Comment 10: Lines 303-305: the sentence is unclear

Response: We clarified the sentences to read the following:

“These findings complement reports of attention-specific activity in ACC^{27,55,56}, supporting the view that ACC plays a role in controlling to which stimulus feature (covert) attention is shifted⁵⁷⁻⁵⁹.” (p. 18)

Comment 11: Figure 8a-b: too many lines and arrows does not allow the reader to easily understand the figure

Response: We've adjusted the figure and corresponding supplementary figure to hopefully facilitate understanding.

Reviewer #2 (Remarks to the Author):

Comment 1: The authors describe a study in which they trained monkeys to learn that one of two colored stimuli was relevant within a block of trials. The animals had to indicate whether a drifting grating within the relevant stimulus was moving up or down, and then make a saccade in the corresponding direction. The color of the relevant stimulus changed at intervals, and the location of the stimulus was irrelevant throughout the experiments. While the animals carried out the task, single neuron activity was recorded from dIPFC, ACC, the caudate and ventral striatum. They found prediction error signals that were modulated by irrelevant feature dimensions, which they referred to as stimulus specific.

This paper addresses an interesting question, about the effects of task-irrelevant stimulus dimensions on prediction error signals across multiple frontal-striatal circuits. The authors have recorded from an impressively large number of neurons, targeting important neural circuitry for this task. The results are clearly presented. I have a number of comments, however, which could help clarify the manuscript.

I was somewhat surprised that the authors did not reference the literature on intra- vs. extradimensional set shifting. This has been extensively studied in prefrontal cortex, primarily by Trevor Robbins group in Cambridge. It seemed relevant to the introduction as it was developed.

Response: We are grateful for the positive remarks about the paper and agree on the importance of the set shifting literature for our task and findings. We therefore added in the revised manuscript more explicit links to the set-shifting literature at various places.

In the introduction we write:

“We did so using a task that employed stimuli that could be characterized by multiple dimensions (color, location, motion), of which however only one was linked to reward outcome across trials (color). Feature values within this reward-relevant dimension were then reversed, akin to intra-dimensional shifts in the set-shifting literature (e.g. Dias et al., 1996; Oh et al., 2014).” (p. 3-4).

In the results section:

“To guide reversal learning towards the goal-relevant color feature value (color 1 vs. color 2), subgroups of neurons should preferentially encode the prediction error for the reward-relevant color dimension. Such color-specific error signals are likely candidates to update the attentional (task-) set following a reversal (Izquierdo et al., 2017)” (p.11)

We also added in the revised manuscript an extended analysis of different reinforcement learning models (in response to reviewer 1) and describe the different model architectures in a new *Supplementary Methods* sections in terms of different assumptions about “attentional sets”.

The added references:

Dias, R., Robbins, T. W. & Roberts, A. C. Dissociation in prefrontal cortex of affective and attentional shifts. *Nature* 380, 69–72 (1996).

Izquierdo, A., Brigman, J.L., Radke, A.K., Rudebeck, P.H., and Holmes, A. (2017). The neural basis of reversal learning: An updated perspective. *Neuroscience* 345, 12-26.

Oh, A., Vidal, J., Taylor, M. J. & Pang, E. W. Neuromagnetic correlates of intra- and extra-dimensional set-shifting. *Brain Cogn.* 86, 90–97 (2014).

Comment 2: That being said, I found the introduction to be a bit out of step with the actual task on which the animals were trained. While the intro frames the paper as being relevant to understanding how the animals learn which features of a stimulus are important, the task only varies the color which should be attended to, to make a choice. Therefore, the animals are not required to learn or switch among different dimensions within the experiment. Clearly they had to learn this during training, but they are not required to do it during the recordings. And, the animals are over-trained, so at this point they are likely only attending to color. It took me a while to work out what the task was, mostly because I had formed a different expectation after reading the introduction.

Response: We recognize the confusion and adjusted, also in response to the other reviewers, the introduction and the rest of the manuscript to better reflect our task design; we clarified the difference between features and dimensions and specified explicitly that all features needed to

be attended to succeed with the task, while color was the only reward-relevant feature dimension.

We adjusted the text at various places to address this (all changes in red font). For instance, in the Introduction, we write in the revised manuscript:

“When faced with novel objects we have to (1) learn which dimensions defining these objects are relevant (e.g. color), and (2) learn the specific feature that is linked to success (e.g. red). To succeed in learning which specific visual features of objects are valuable likely depends on estimating feature relevance and improving this estimate through trial and error learning¹⁻³.” (p. 2)

“Specifically, we asked (1) whether prediction error signals in these regions could be informative about the specific features (e.g., upwards motion, color red, etc.) that were just chosen and presumably led to the unexpected outcome, and (2) whether such feature-choice informative prediction errors would occur more commonly for the reward-relevant dimension (e.g. color) as opposed to reward-irrelevant dimensions (e.g. motion).” (p. 3)

In the results, we specify:

“This task required (1) feature-based attentional selection of one over another stimulus based on a reward associated color, (2) to use the motion direction of the attended stimulus to program a saccadic response, and (3) to make a response only when the attended stimulus dimmed. Therefore, stimulus location (spatial attention) and stimulus motion-direction were relevant to the response on a trial-by-trial basis, while only stimulus color was linked to reward across trials.” (p. 5).

In the discussion, we clarified:

“Furthermore, following covert attentional selection, spatial and feature-based components of attention most likely modulate the activity of non-color selective neurons to solve the task. In particular, attentional modulations of neurons encoding motion direction (upwards vs. downwards) with the proper receptive fields may allow acting according to the relevant sensory-motor association (upwards vs. downwards saccade).” (p. 19).

Comment 3: Several studies have shown that prefrontal cortex (mostly dlPFC) contains a representation of task irrelevant stimulus dimensions (for example Constantinidis and Goldman-Rakic, as well as Mante, Sussillo and Newsome). This is also likely related to mixed selectivity, although it is possible to have one without the other.

Response: We agree and thank the reviewer for suggesting this link. We added explicit links to the mixed selectivity literature, as well as literature showing encoding of non-relevant task dimensions in the discussion. We write in the revised manuscript:

“However, our findings also clarify that specific information about the origin of prediction errors, with regards to the currently rewarded feature, is not localized to the ACC, but widely distributed to all areas we recorded from and which are known to be anatomically mono-synaptically connected^{2,16,60-63}, and functionally synchronized in different task contexts^{55,56,64,65}. Rather than locally segregated encoding of prediction errors and feature choice information, the distributed nature of feature-specific prediction error signaling we report here supports recent summaries of the recurrent nature of fronto-striatal processes

underlying reward-based choices⁶⁶. In contrast to previous views that imply more localized and serial computations of subprocesses of goal-directed choices^{67,68}, these recent conceptualizations propose that many brain areas act in concert to perform similar computations, that these distributed networks are highly recurrent, and that such distributed recurrent networks are organized into functional and temporal hierarchies^{28,59,66,69}. This additionally fits the notion that prediction error signals seem to be encoded in many brain regions and in an almost redundant fashion^{70,71}. In line with our findings, many neurons, especially in frontal regions, have previously been shown to encode a mixed selectivity of task variables^{72,73} (see also **Figure S3d**). Further related to this, we found substantial amounts of neurons that encoded prediction errors selectively for non-reward relevant stimulus dimensions (motion/location). It has previously been shown that non-relevant task variables and features are represented and tracked over multiple trials in the brain, in particular in prefrontal regions^{58,74-76}, and it has been argued that the encoding of non-relevant information may promote behavioral flexibility in volatile task environments by facilitating the detection of changes in this environment^{76,77}. ” (p. 18-19)

Comment 4: It should be made clear that rewards were deterministic, when the task is described at the beginning of the results. While this is a learning task, the paradigm is more consistent with the cognitive literature than the reinforcement learning literature.

Response: We added in the results, discussion and methods section explicit statements about the deterministic reward schedule (p. 5, p. 27).

We agree that our paradigm is consistent with the cognitive literature, but show with added model evaluations in the revised manuscript that the key mechanism of reinforcement learning (prediction error based updating) is important to understand the learning in our task. In particular we tested how well a Bayesian learning model accounts for the monkeys' reversal learning performance. This Bayesian learning does not use prediction errors, but rather tracks reward probabilities of stimulus features. This model performed worse in reproducing behavioral performance than four reinforcement learning models (new Figure S1), suggesting that prediction errors are important to understand the reversal learning task. We added information about this finding in the revised text.

We also think that the choice of the reward schedule (deterministic versus probabilistic) does not affect whether choices are based on value (which is a key proposition in reinforcement learning), but rather influences the difficulty and reward uncertainty of learning.

We hope that by adding more references to the set shifting literature (comment 1, above), we made the link to the cognitive literature more explicit in the revised manuscript.

Comment 5: The reward outcome is largely confounded with the choice in this task. I guess they can be dissociated after reversals. Is there a clear temporal signature of this?

Response: We are not entirely certain we fully understand the reviewers' concern or question, but address a few possibilities:

- *In our task, the choice determines the reward outcome because of the deterministic reward schedule. However, choice and reward delivery were temporally separated by*

approximately 450 ms. This time separation between the two epochs allows discerning the neuronal responses associated with these cognitive processes. For instance, by the time the outcome was revealed, no stimuli were displayed on the screen. Thus, sensory modulations of neuronal responses, which are important in decision making (i.e., choice), do not interfere with reward processing.

- Choices in this task were linked to the particular stimulus and its features that the animal responded to. Since the values of the colored stimuli alternated in this reversal learning task, color choices were distinct from reward outcomes across a session. Location and motion direction choices were unrelated (uncorrelated) to reward outcomes at all times.
- For all the RPE signals we identified (positive, negative, unsigned) in this manuscript, reward was either constant (positive, negative RPE) or specifically partialled out as a covariate (unsigned RPE), which means that the feature-specific RPE encoding neurons we identified were independent of the outcome (rewarded or unrewarded).

Comment 6: I was not clear about why positive and negative reward prediction errors were separated. Why not just assess whether neurons encoded either signed or unsigned RPEs? Also, perhaps the neurons were encoding chosen value and reward, instead of RPE, since RPE can be calculated from these two quantities. This should be assessed.

Response:

1. We chose to split signed prediction errors into positive and negative RPEs for several reasons: (a) it allowed us to obtain information about the tendency to carry feature-information in each of the two trial types (correct / error trials) and prediction errors. (b) Across the literature, positive and negative RPEs are often dissociated (e.g. Kennerley et al., 2011; Asaad and Eskandar, 2011; Matsumoto et al., 2007; Stalnaker et al., 2012) and (c) it is not necessarily warranted to assume that the same neurons linearly encode positive and negative prediction errors. It has previously been observed that populations of neurons encoding positive and negative RPEs only partially overlap (e.g. Kennerley et al., 2011). (d) In line with this, there is some evidence that animals may learn differently from positive and negative outcomes (Matsumoto et al., 2007; Fusi et al., 2007; Asaad and Eskandar, 2011). (e) For our main analysis, and in line with previous research, we identified neurons encoding a negative prediction error as those that increased their firing rate with greater (more negative) RPEs (although we now add the opposite definition in response to Reviewer 1, new Figure S5). This definition would not be in line with a linear encoding across positive and negative RPEs. (f) Lastly, we combined correct and error trials to identify neurons that encoded an unsigned RPE.
2. With regards to the possibility that neurons were instead encoding chosen value (V) and reward (R) instead of RPE during the reward outcome period: Because RPE is computed as $RPE = (R - V)$, and because R is 0 for error trials and 1 for correct trials, RPE and chosen value are anti-correlated (R -value = -1) in both correct and error trials. Furthermore, as discussed above, we analyzed positive and negative RPEs separately, which means reward was constant in either group. For the surprise signals that contained both correct and error trials, reward was partialled out using a co-variate vector of ± 1 for correct/error trials that was included in the original partial correlation analysis of firing rate with RPE value. For this reason, surprise and choice value were also anti-correlated with an R -value of -1. This means that every neuron we identified as correlating with a positive, negative, or unsigned RPE signal was by default also correlated (in the opposite direction) with the value of the chosen stimulus.

We have added text to the manuscript to clarify that we cannot distinguish these two types of variable encoding:

“Please note that RPE is computed as $RPE = R - V$ (where R denotes reward, 1 or 0, and V denotes expected value of the chosen stimulus), indicating that RPE at the time of reward is anti-correlated with the value of the chosen stimulus prior to the time of reward, making RPE and ‘chosen-value’ non-independent variables that are difficult to distinguish.” (p. 8)

3. In addition, our manuscript provides direct information how reward outcome (R) is encoded in general (Fig. S3) and we quantify how reward outcomes are encoded with regard to specific features (new Figure S8). This later analysis showed that prediction error encoding is realized by a neuronal population that is largely segregated from neurons encoding the reward outcome (rewarded/non-rewarded) per se. We write in the revised text:

“Feature-specific correlations of firing rates with the RPE signals might be evident in populations of neurons that show already feature-specific firing for different outcomes (rewarded vs. non-rewarded) irrespective of prediction error, or they could occur in a segregated neuronal population. To answer this question, we first calculated the prevalence of feature information in the outcome period, by testing whether neurons encoded any of nine feature-related variables using multiple regression analysis (see **Suppl. Methods**). Since neurons likely encoded more than one variable (**Figure S3c, d**), we evaluated feature information encoding at the first order (greatest regression coefficient) as well as at the second order, in which case a bi-linear regression with two variables fit a given neuron significantly better (see **Suppl. Methods**). We found that 20/16/13% of neurons encoded color/motion/location specific information about the outcome at the first or second order. However, only 35/28/26% of those neurons also showed significant color/motion/location firing correlations with RPEs. This suggests that a substantial population of feature-specific RPE encoding neurons cannot be explained by prediction error-independent feature tuning during the outcome period of the task (**Figure S8**).” (p. 13-14)

Comment 7: The results in Fig. 3 were unconvincing. Such a small number of neurons encoded non-specific RPEs that it seems to be an irrelevant feature of the neural response. Comparison of onset timing of this feature with timing of feature specific RPEs does not seem meaningful.

Response: We understand that the relatively small number of neurons that solely encode non-specific RPEs may seem unconvincing, but we want to point the reviewer to the specification on page 10-11:

“These results were quantitatively and qualitatively similar when we allowed individual neurons to be part of both groups of non-specific and feature-specific RPEs (data not shown; Kolmogorov-Smirnoff test $p < .0001$; Rank sum test $p < .0001$; $n_{\text{feat-spec}} = 774$; $n_{\text{non-spec}} = 587$), [...]”.

For this analysis, we did not restrict non-specific RPE neurons to those neurons that solely encode a non-specific RPE, but included those neurons that could be identified as both (with

potentially different time courses). The results were equivalent with this larger population of non-specific RPE neurons. We opted to display/report the main results comparing the small 'true' non-specific RPE population with the feature-specific RPE population, as these populations were non-overlapping and it is less meaningful to identify a neuron as encoding a non-specific **and** feature-specific RPE.

Furthermore, we added new analyses (in response to Reviewer 1) in which we increased the threshold of identifying RPE neurons from 4 consecutive significant bins to 6 consecutive significant bins; we found that the latency difference between non-specific and feature-specific RPE signals is further increased with this more conservative approach (Figure S4a).

To allow the reader to more readily see how many neurons were included in the analyses, we added specific information about the total numbers of neurons used in the revised main text and figure legends.

Comment 8: What p-value was used to assess significance for Fig. 3?

Response: We convey on page 10 and 32 in the revised text that we employed three different statistical tests to evaluate the time course differences between feature-specific and non-specific RPEs. We compared the cumulative sums of the distributions (Figure 4a, b) using a Kolmogorov-Smirnov test that was multiple comparison corrected (Bonferroni-Holm method), and additionally verified results with a non-parametric ranksum test that was also multiple comparison corrected. Finally, we determined the time point at which 25% of RPE encoding occurred and compared these time points using a randomization procedure (number iterations = 500).

Comment 9: I would refer to PFC as dlPFC, since prefrontal cortex is usually defined broadly as OFC, ACC, dlPFC, vlPFC, etc.

Response: We've adjusted the terminology accordingly.

Comment 10: It seems there may be a firing rate confound in the cell-type analysis. High firing rate neurons encode more information than low firing rate neurons. What if firing rate is controlled for in the analysis which compares coding in excitatory and inhibitory neurons?

Response: We agree that this could be of concern and should be quantified explicitly. To this end we performed the following new analyses for the revised manuscript:

- a) *When setting an arbitrary firing rate threshold of 10Hz in order to remove interneurons with very high firing rates, the ratio of putative interneurons to putative principal cells that encode a feature-specific RPE remains greater than the ratio of interneurons to principal cells observed in the population (Chi-square test, $p = 0.048$). We report this as follows:*

"Since it is possible that the previously described effects are mainly driven by narrow spiking neurons with a high firing rate, we performed several control analyses. First, we repeated the analysis by limiting neurons to a maximum firing rate of 10Hz. We found equivalent results, whereby the ratio of narrow spiking to broad spiking cells that encoded feature-specific RPEs was greater than the ratio

observed in the population in frontal regions (Chi-square test, $p = 0.048$), and as a trend in striatal regions (Chi-square test, $p = 0.09$), while this was not the case for non-specific RPE encoding neurons (Chi-square test, both $p > 0.35$). (p. 15)

- b) Additionally, to increase the sample size we combined the populations from frontal and striatal regions, since in our main analysis we observed a similar trend in the striatum as in frontal regions, whereby feature-specific RPEs were somewhat more likely to be encoded by narrow spiking neurons (chi-square test, $p=0.08$). With the combined populations, to test a relatively narrow firing rate window, we tested only those neurons with a firing rate between 0.5 and 5 Hz, and observed the same result – the ratio of narrow to broad spiking cells encoding feature-specific RPEs was greater than the ratio of narrow to broad spiking cells observed in general (Chi-square test, $p = 0.014$). We report this as follows:

“Secondly, when we combined frontal and striatal neurons to increase the sampling size, the ratio of narrow spiking to broad cells that encoded feature-specific RPEs was greater than the ratio of the general population even when firing rates were restricted to 0.5 - 5Hz (Chi-square test, $p = 0.014$), while this was not the case for non-specific RPE encoding neurons (Chi-square test, $p = 0.42$).” (p. 15)

- c) When we arbitrarily bin neurons into multiple bins according to their firing rates, the sampling rates in each bin become too small for reliable statistical analysis; however, the ratio of narrow spiking to broad spiking cells that encode a feature-specific RPE remains greater than that observed in the population in almost every bin. This suggests that the effect we observe is not only driven by high firing interneurons but seemingly occurs across a wider range of firing rates. We report this in the new Supplementary Figure S9 and in the text as follows:

“Lastly, when arbitrarily binning neurons in multiple bins according to their firing rates, the ratio of narrow spiking to broad spiking cells is greater for feature-specific RPE encoding neurons than for the general population in most firing rate bins (**Figure S9**)” (p. 15)

Reviewer #3 (Remarks to the Author):

Oemisch et al. report in this manuscript findings from a study of feature-specific reward prediction error (RPE) signals in four different brain areas (two cortical: ACC & LPFC, two striatal: VS & CD). The topic of the study is important and timely, and the experimental effort is exceptional (1960 units recorded in 4 different brain areas in two monkeys). However, I have a few comments and questions that should be addressed.

Comment 1: Main concerns:

1.) I do not fully understand the two hypothesis that are tested in this paper. Describing them more clearly and contrasting the findings in this study with the previous one in PFC would be very helpful to understand the larger significance of the paper.

Response: We are now more explicit in outlining our hypotheses, specific goals and the corresponding findings.

We adjusted parts of the introduction and write in the revised manuscript:

- *To specify the questions we asked:*

“A prediction of these behavioral models is that brain circuits need to combine information about the occurrence of a prediction error with information about the specific stimulus feature of the relevant dimension that should be attended in future trials¹⁵.” (p. 3)

and

“Specifically, we asked (1) whether prediction error signals in these regions could be informative about the specific features (e.g., upwards motion, color red, etc.) that were just chosen and presumably led to the unexpected outcome, and (2) whether such feature-choice informative prediction errors would occur more commonly for the reward-relevant dimension (e.g. color) as opposed to reward-irrelevant dimensions (e.g. motion).” (p. 3)

- *to better emphasize the two hypotheses with regards to local contributions:*

“In one scenario emphasizing functional localization, a general prediction error signal may emerge locally within the ventral striatum and is broadcasted to prefrontal cortex where it modifies the activity of feature selective neurons¹⁵. Updated prefrontal cortex neurons might then exert an improved top-down signal over sensory cortices for attention and choices in subsequent trials²⁰⁻²². This view is supported by mostly human fMRI findings that single out the striatum as core region to encode prediction errors^{23,24}, and the lateral prefrontal cortex to encode a feature-based top-down signal^{4,25-27} together with prediction error information¹⁵. In contrast to this scenario that emphasizes functional localization, neurons encoding prediction errors might be distributed beyond the ventral striatum and carry early on explicit feature choice information in multiple areas²⁸. Activity of such neurons could serve as a feature-specific eligibility trace²⁹, orchestrated across the recurrent fronto-striatal loops. Such a distributed, feature-specific eligibility trace is predicted by network models that learn relevant features by using attentional feedback signals to label synapses among those neurons that also contributed to the feature-specific reward prediction itself^{30,31}. (p. 4)

and parts of the discussion:

- *To reflect which hypotheses regarding functional localization our results of feature-specific prediction errors support:*

*“However, our findings also clarify that specific information about the origin of prediction errors, with regards to the currently rewarded feature, is not localized to the ACC, but widely distributed to all areas we recorded from and which are known to be anatomically mono-synaptically connected^{2,16,60–63}, and functionally synchronized in different task contexts^{55,56,64,65}. Rather than locally segregated encoding of prediction errors and feature choice information, the distributed nature of feature-specific prediction error signaling we report here supports recent summaries of the recurrent nature of fronto-striatal processes underlying reward-based choices⁶⁶. In contrast to previous views that imply more localized and serial computations of subprocesses of goal-directed choices^{67,68}, these recent conceptualizations propose that many brain areas act in concert to perform similar computations, that these distributed networks are highly recurrent, and that such distributed recurrent networks are organized into functional and temporal hierarchies^{28,59,66,69}. This additionally fits the notion that prediction error signals seem to be encoded in many brain regions and in an almost redundant fashion^{70,71}. In line with our findings, many neurons, especially in frontal regions, have previously been shown to encode a mixed selectivity of task variables^{72,73} (see also **Figure S3d**).”* (p. 18-19)

- in addition to previously already highlighting how these results may support a recent model (Rombouts, J. O., Bohte, S. M. & Roelfsema, P. R. How Attention Can Create Synaptic Tags for the Learning of Working Memories in Sequential Tasks. *PLoS Comput. Biol.* 11, 1–34 (2015).):

“The models make multiple predictions that are supported by our data. Firstly, synaptic updating is taking place in an associative network layer that resembles the fronto-striatal network of value learning as opposed to sensory or motor related network layers. Secondly, feature-specific prediction errors are predicted to emerge as local neuronal signals across the entire associative network based on neuron-specific synaptic tags, closely corresponding to the distributed RPEs we observed. Finally, the model predicts that a learning of task relevant features depends on attention towards those stimulus features that are most consistently reward associated. This attentional hypothesis of reinforcement learning was directly tested in our experiment, providing evidence that the most ubiquitously encoded prediction error signals occur for the attended, goal-relevant color feature.” (p. 23-24)

Comment 2: I do not completely understand the task. Specifically, the description of the task in the methods section does not completely fit what is shown in figure 1a. For example, why do the gratings change color to black/green and black/red (for one monkey, different colors for the other monkey)? Why not red/green? If one of the gratings is black, it is obvious, which grating should be followed. That reduces the effort of allocating attention correctly. More importantly, I do not understand the section coming after that (starting on page 21) that describes a complex sequence of changes in the displays. I do not understand what is going on and in which temporal order. A comprehensive figure might help. More importantly, I do not understand the rationale for this task design. Lastly, I do not understand why the authors use the complex dimming pattern in their task. Again, what is the rationale?

Response: There was a clear rationale for each of the components of the task which we describe explicitly in the revised text. We apologize for the lack of clarity in the original text and adjusted text and figures of the revised submission.

- *We originally specified the grating colors as black/red and black/green in the text to acknowledge that the stimuli were not homogeneously red or green but always contained black bars (moving within the stimulus aperture). The reviewer is correct – at a given time, one stimulus was red (with black bars) and one stimulus was green (with black bars) – there was never an instance in which one stimulus was e.g. red and the other black. We clarified this in the text by omitting the highlighting of the black bars:*

“After 0.5 - 0.9s, two black/white gratings appeared to the left and right of the central fixation point. Following another 0.4s the two stimulus gratings either changed color to green and red (monkey K: cyan and yellow), or they started moving in opposite directions up and down, [...]” (p. 26)

- *The task controls covert attentional allocation to one of two stimuli by varying the dimming events temporally between the two stimuli. A dimming event constituted the Go-signal for the animal to make a response. Specifically, if the animal chose to respond to the red stimulus, it had to make that response at the time the red stimulus dimmed – to ascertain continuous allocation of attention to the red stimulus, dimming of the red stimulus could occur at different (relative) time points – either before the green stimulus dimmed, at the same time as the green stimulus dimmed, or afterwards, in which case the animal had to ignore dimming of the green stimulus and wait for dimming of the red stimulus. We adjusted the figure to enhance understanding and clarified this in the text as follows:*

“The rationale for this design was to ascertain continuous allocation of attention to one stimulus - since the animal did not know the time of dimming of the current target stimulus (which could occur before, after, or at the same time as the second stimulus), it had to attend continuously until the ‘Go-signal’ (dimming) of that stimulus occurred. If dimming of the target stimulus occurred after dimming of the second/distractor stimulus, the animal had to ignore dimming of the second stimulus and wait for dimming of the target stimulus. “(p. 26-27)

- *The task allowed inferring which stimulus was attended prior to the choice. The motion direction of the two stimuli were never the same in a single trial. This allowed us to infer to which stimulus the animal attended, because in the one third of trials with simultaneous dimming of both stimuli the animal made as often a saccadic response in the direction of motion of the high rewarded stimulus as in trials when the dimming happened only in the high rewarded stimulus.*

Comment 3: Since the reversals were only in the color domain, the monkeys should have learned to completely ignore the other stimulus features (location, motion). Was that the case? The authors never describe the final best-fitting model and what the relative weight of the different feature dimensions is. This would be interesting in two ways. First, if the monkeys still responded to RPE in the non-relevant dimensions, one has to ask why the monkey was unable to learn to ignore them. Second, if the monkey learned to ignore them, the question arises, why there were feature-specific RPE signals for non-relevant dimensions.

Response: We appreciate the comment and added various lines of information to the revised text (1) quantifying that location and motion had a residual influence, and (2) conveying why the task design made it difficult to ignore motion and location information (and thus made credit assignment difficult). These changes included the following:

- (a) *Although location and motion were stimulus dimensions that were not systematically linked to reward across trials, both remained relevant to the response on a trial-by-trial basis, which could potentially explain why these dimensions, albeit to a lesser degree, were still represented by feature-specific RPE neuron populations:*

“This task required (1) feature-based attentional selection of one over another stimulus based on a reward associated color, (2) to use the motion direction of the attended stimulus to program a saccadic response, and (3) to make a response only when the attended stimulus dimmed. Therefore, stimulus location (spatial attention) and stimulus motion-direction were relevant to the response on a trial-by-trial basis, while only stimulus color was linked to reward across trials.” (p. 5)

These requirements made the task particularly difficult for the monkeys (their asymptotic performance was only around 75%, see Fig. 2a) when compared to other complex and cued attention tasks we have used before, and also compared to tasks in which the stimulus location coincides with the action goal location (e.g. making a saccade directly to the attended stimulus, as e.g. in Lau & Glimcher 2008, Neuron).

*To describe more explicitly the best-fitting (and possible alternative) models we evaluated for the revised manuscript five different reinforcement learning models to convey explicitly that a color-only model was not the model best explaining the monkeys performance. Rather the dimension-weighting component of the best-fitting model contributed to better explaining the learning performance even after accounting for the additional (weighting) parameter that was used in that model using the Akaike Information Criterion. The model evaluations are shown in a newly added **Supplementary Figure S1**. In our (cross validation) approach the “best fitting” model is the architecture that predicts the choices of the subject, the correct and incorrect ones. The benchmark is the log-likelihood of the responses given the model.*

Please see the reply to comment 4 of reviewer 1 for a more extensive overview of the remaining models.

We added a more detailed description of the different tested models to the revised text (with model descriptions in a new Suppl. Methods section). We write:

*“In order to discern trial-by-trial neuronal encoding of reward prediction errors (RPEs) we evaluated various reinforcement learning (RL) models using RPEs to account for feature-based learning performance^{5,13} (**Suppl. Methods**). The models used RPEs to update values either (1) for all stimulus features independently, (2) for only the different colors that were systematically linked to reward, or (3) for all features but weighted by a dimensional weight. In addition, we evaluated a Bayesian learning model that tracked reward expectations for each stimulus feature without the use of RPEs. We found that the monkey’s learning performance was best explained by the model that learns to weight values of specific features by their dimension, i.e. that learned a location, motion direction and color weight, and additionally decayed feature-specific values of the non-*

chosen stimulus. These findings corroborate previous studies^{5,13} suggesting that both dimension and feature information contribute to learning performance (**Figure 2b, Figure S1, Suppl. Methods**). The Dimensional-weighting of features plus decay model was superior to models that either treated all features non-selectively, or that only included the color dimension by artificially removing the influence of the location and motion dimension. This was evident for both monkeys in lowest Akaike Information Criteria scores across models and low cross-validation log-likelihood values (**Figure S1**). We thus used that model to generate trial-by-trial estimates of RPEs as shown in **Figure 2c**, allowing us to correlate neuronal firing rates during the -0.5-1.5 sec. reward outcome epoch of each trial time-resolved in 200ms windows shifted by 50ms (**Figure 2d, e**).” (p. 7)

- (b) We added text describing more explicitly that the animals did not switch from one color choice to the second color choice within one trial, but instead on average took several trials to switch from one color choice to the other (Fig. 2a). Furthermore, the asymptotic average performance in both animals is reached at around 75% correct choices, far from optimal, highlighting the general difficulty of the task, which was likely driven by the need to integrate multiple stimulus features into a single response:

“Even though reward was deterministic, **Figure 2a** highlights the deviation from optimal performance in both animals – choice switches from one color to the other did not occur within a single trial and the asymptotic average performance was reached around 75% correct choices. This illustrates the general difficulty of the task, which was likely driven by the need to integrate multiple stimulus features into a single response.” (p. 6)

- (c) In addition, we added a discussion section devoted to this topic. It is commonly observed that neurons, especially in frontal regions, encode irrelevant task variables, and/or a mixed selectivity of multiple task variables, and that this may be computationally advantageous to adapt to unexpected changes in the environment:

“This additionally fits the notion that prediction error signals seem to be encoded in many brain regions and in an almost redundant fashion^{70,71}. In line with our findings, many neurons, especially in frontal regions, have previously been shown to encode a mixed selectivity of task variables^{72,73} (see also **Figure S3d**). Further related to this, we found substantial amounts of neurons that encoded prediction errors selectively for non-reward relevant stimulus dimensions (motion/location). It has previously been shown that non-relevant task variables and features are represented and tracked over multiple trials in the brain, in particular in prefrontal regions^{58,74–76}, and it has been argued that the encoding of non-relevant information may promote behavioral flexibility in volatile task environments by facilitating the detection of changes in this environment^{76,77}. Furthermore, following covert attentional selection, spatial and feature-based components of attention most likely modulate the activity of non-color selective neurons to solve the task. In particular, attentional modulations of neurons encoding motion direction (upwards vs. downwards) with the proper receptive fields may allow acting according to the relevant sensory-motor association (upwards vs. downwards saccade).” (p. 19)

Comment 4: The authors should include a table with the exact number neurons that carry RPE signals for a specific feature.

Response: We added a new Supplementary Table 1 outlining the numbers (and percentages) of neurons encoding the different types of feature-specific RPEs.

Comment 5: More importantly, it is not clear to me how specific a feature-specific RPE signal really is. Do these neurons only carry RPE signals for one feature dimension (and no other)? Within a feature dimension, are the neurons specific for only one out of two possible types (as implied in Fig. 2) or is there more variety. Asked another way, are the feature specific and unspecific RPE signals two very different, non-overlapping classes or are they along a spectrum, where some cells are very specific, some cells are specific for a few features and some cells for all features? The authors need to better document their results in that regard.

Response: We thank the reviewer for the question. Across feature dimensions, a neuron can be identified as encoding more than one feature-specific RPE signal (e.g. a color-specific and location-specific RPE). Within a feature dimension, RPE encoding had to be specific to one feature (e.g. color 1), otherwise it was considered non-specific. This was determined by the correlation between FR and RPE, which needed to be stronger for one feature value (e.g. color 1) than the second feature value (e.g. color 2). This means that a given neuron could either a) encode a significant RPE for color 1 and color 2 but do so significantly stronger for one of the two, or b) encode a significant RPE for only one color and not the other. Our statistical analyses did not differentiate between these two cases. Lastly, neurons could technically be identified as encoding both, a non-specific and a feature-specific RPE – consider the previous example where a neuron significantly encodes an RPE signal following color 1 choices and color 2 choices, however significantly more strongly for color 1; this neuron would still ‘show up’ statistically as a non-specific RPE encoding neuron when color 1 and color 2 choices are collapsed. Since it is not meaningful to label a neuron as feature-specific and non-specific, we considered a neuron only as a non-specific RPE neuron if it could not also be identified as a feature-specific RPE encoding neuron, making these two separate populations. We previously only briefly addressed this, but now added in the revised manuscript a more specific statement in the Methods section:

*“Note that technically a neuron could be identified as encoding a non-specific and feature-specific RPE. Consider the following example: A neuron may encode a significant RPE for color 1 and color 2 choices, but does so significantly stronger for color 1 choices. This neuron would statistically still ‘show up’ as a non-specific RPE encoding neuron when color 1 and color 2 choices are collapsed. Since it is not meaningful to label a neuron as feature-specific **and** non-specific, for any analysis that explicitly compared feature-specific with non-specific RPEs (e.g. Figure 4, Figure 8), a neuron was only considered as a non-specific RPE neuron if it could not also be identified as a feature-specific RPE neuron, making these two separate populations.” (p. 31-32)*

*To address the reviewer’s suggestion to provide more explicit results about the spectrum of neuronal encoding between feature dimensions and trial types, we added a new **Supplementary Figure S6** to highlight the overlap in neuronal populations involved.*

Related to this comment, we added to the discussion of the revised text a section describing that our results are consistent with a mixed selectivity coding of prediction errors (more details in reply to Comment 3 of reviewer 1).

Comment 6: The meaning of ‘negative’ and ‘positive’ RPE is slightly unclear. Is it a neg/pos correlation of firing rate with RPE along its entire axis, or neurons that encode only negative or positive deviations from expectations (but not deviations with the other sign)?

Response: We apologize for the confusion. By our definition, and similar to previous studies (e.g. Kennerley et al., 2011), a negative RPE occurred following incorrect (i.e. non-rewarded) trials only, while a positive RPE occurred following correct (i.e. rewarded) trials only. We clarified this in the revised text as follows:

“By definition, positive RPEs (pRPE) occurred solely on correct trials, while nRPEs occurred solely on error trials.” (p. 8)

Comment 7: The idea that feature-specific RPE signals follow unspecific ones is not supported by the data. In the earliest time bins there are already a substantial number of feature-specific neurons. The authors need to be more careful with their conclusions.

*Response: We acknowledge this point and tone down our wording and conclusions in the revised text to reflect better the actual statistics. Overall, we agree that the time bin distributions of feature- and non-specific RPE encoding neurons are overlapping, but find it still meaningful to compare the means of these distributions. We do this using three separate statistical tests (see p10+32, Kolmogorov-Smirnoff test, Rank sum test, randomization procedure), and find that on average feature-specific RPEs are encoded later. Furthermore, with a more conservative approach in response to Reviewer 1, this difference in latency becomes more pronounced (newly added **Figure S4**). In response to the reviewer we nevertheless tone down our conclusions to read the following:*

*“The effect was partly driven by feature-specific RPE encoding neurons showing a continued increase and remaining at a higher plateau level for a longer duration than non-specific RPE signals (**Figure 4a, b, Figure S4a**). Thus, while some individual neurons did show early feature-selective RPE encoding, at the population level, the set of neurons encoding feature-specific error information showed a slower time course, which was sustained at a higher level compared to non-specific error information.” (p. 10)*

Comment 8: Lastly, the firing pattern dynamic that the authors interpret as evidence for reinforcement learning in the attention domain is so unspecific that most learning processes would produce it. This hypothesis is not convincingly supported and should be taken out of the paper.

Response: We believe there might be some confusion we introduced with some wording and try to address this by clarifying the results and being more careful in interpreting them in the revised text.

In particular, the firing rate dynamics the reviewer refers to are specific to the preferred color of the neurons encoding color-specific prediction errors: These neurons show a differential increase in firing at the next trial’s color onset (trial $n+1$) only if the current choice (trial n) was to the preferred color of the neurons and not when it was to the non-preferred color of the neurons. We summarize this by saying that:

“This increased color onset response was on average stronger following trials with low RPE than high RPE, i.e. after learning (Figure 2d, Figure 9a, b, Figure S10), specifically when the preceding trial’s choice was for the preferred color of the neuron, i.e. the color for which it selectively encoded a greater RPE signal (Figure 9a, b cyan versus grey bars, Figure 3 top panels).” (p. 16)

We agree that various learning models would be able to generate this response, but it remains a response that is tied to the feature that is preferred by the neurons (in terms of RPE encoding).

To address the reviewers concern, we adjusted the text to align our interpretation more specifically to the actual results we report by writing:

“These findings provide evidence that color-specific prediction error signals during the early learning trials after reversal may translate into color cue firing rate increases for these same neurons after learning has taken place, potentially reminiscent of the temporal transfer of classical dopaminergic prediction error signals.” (p. 17)

Reviewers' comments:

Reviewer #1 (Remarks to the Author):

[This reviewer had to withdraw from the review process, but provided comments via email to the editor requesting additional clarification. The substance of these requests was provided to the authors.]

Reviewer #2 (Remarks to the Author):

The authors have addressed most of my concerns. However, I have a few remaining comments.

Comment 1: With respect to this statement: "Please note that RPE is computed as $RPE = R - V$ (where R denotes reward, 1 or 0, and V denotes expected value of the chosen stimulus), indicating that RPE at the time of reward is anti-correlated with the value of the chosen stimulus prior to the time of reward, making RPE and 'chosen-value' non-independent variables that are difficult to distinguish."

This will not be true during reversal learning. Because values change so they are not always 0/1. This should be clarified and addressed.

Comment 2: The Bayesian model was confusing. I was expecting that the authors would use a binomial distribution to represent the reward probabilities associated with each color. Bayesian models are problematic with deterministic outcomes, because there is no way to model the deterministic mappings. I could not really sort out what f was in the development of this model? How was p_r calculated?

Comment 3: I understand that the stimuli the animals were choosing did in fact have additional features other than color. But the values of these features should never go above 0 on average. So I am still not clear why the authors think that there will be any feature specific prediction errors about these dimensions. This was not clarified in the introduction. The task used does not allow the authors to address these issues, unless it can be shown that the animals did tend to try to use these other dimensions for learning. But then this raises the question of why they would do so, since those dimensions did not predict reward.

Reviewer #3 (Remarks to the Author):

The authors have made many improvements in the manuscript that have greatly clarified their findings. The overall conceptual design of the experiment and the meaning of the findings is now clear. I have no further questions. I think this is an interesting contribution and deserves publication in Nat. Communication.

Point-by-point Reply - Summary of Changes in the Revised Text:

We are grateful that reviewer 2 and reviewer 3 found that our revised manuscript clarified most (reviewer 2) or all (reviewer 3) of their comments and improved the manuscript. We also acknowledge reviewer 1's request to make more accessible to readers the variations of feature specific prediction errors.

Our revision addresses these requests from reviewer 1 and 2 explicitly and constructively. All our findings and interpretations remained valid.

In summary, our revisions provide the clarifications that reviewer 1 and 2 requested about how our best-fitting Bayesian-Reinforcement Learning model works and how it relates to some of our findings. We describe these clarifications in detail in the point-by-point reply and summarize them here:

- We provide clarifications of how reward prediction errors (RPEs) are computed. Reviewer 2 detected a possible misunderstanding about this that we try to prevent with improved descriptions about RPE calculations.
- We describe to Reviewer 2 how the Bayesian weighting of feature values is implemented in our model and in previous papers. There was a misunderstanding that we believe is prevented by referring to a more extensive methods section in the revised manuscript.
- We address Reviewer 1's comments by adding a Supplementary figure (Fig. S13) showing for each monkey the prediction errors separately for individual features. This figure is what can be expected from the reward prediction error plots we are showing in Fig. 2 of the main text.
- We carefully read the manuscript text, the supplementary text and the figures and made few minor corrections to grammar/spelling. We also adjusted supplementary figure numbering.

All changes to the main text are highlighted in red font of the submitted text.

We would be grateful if you considered our revised manuscript suitable for publication in *Nature Communications*.

Thank you very much.

Sincerely,

Mariann Oemisch
Stephanie Westendorff
Marzyeh Azimi
Ali Hassani
Salva Ardid
Paul Tiesinga
Thilo Womelsdorf
[the authors]

Reviewer #1 (Remarks to the Author):

Comment 1: [The reviewer requested clarification of feature-specific prediction errors.]

Response: We follow the reviewer's specific suggestion and show in a new Supplementary **Fig. S13** the prediction errors calculated for individual features of the stimulus the monkeys chose. These prediction errors denote the difference of the feature-specific value minus the outcome ($RPE_{\text{feature}} = \text{Outcome} - V_{\text{feature}}$). We added the reference to the new Supplementary **Figure S13** by writing:

"The trial-by-trial development of the average positive and negative RPEs for each monkey are shown in **Fig. 2d**. We also illustrate the prediction errors for each of the features of the chosen stimulus separately, confirming that the RPE's used by the model to adjust feature values are dominated by the color dimension, while the reward-irrelevant motion and location dimensions show a similar progression of RPE but at a lower magnitude corresponding to their lower feature values (**Fig. S13**)." (p. 29)

We show the feature-specific encoding of prediction errors in multiple examples in Fig. 2 (over trials) and Fig. 3 (over different conditions) and describe these examples in the results section in detail.

Comment 2: [The reviewer asked why monkeys had a tendency to switch their choices after location and motion direction; and whether this was consistent with the model.]

Response: The behavioral effects are in agreement with the model, and we describe the reasons why the best fitting model captures the influences of motion and location in addition to the more prominent influence of the color dimension.

We agree that this makes a strong case.

The best fitting model deploys a dimension weight that weights the values for individual features. This "*Feature-Dimension weighted-Decay RL model*" learns which of the features is most reward predictive. The model thus explicitly considers that location and motion direction influences the monkeys' choices in addition to the more prominent influence of the color feature.

We exclude the two alternatives to this model. A model that only considers influences from color is not as good as the "*Feature-Dimension weighted-Decay RL model*". A model that treats all features identical and non-selectively, also does worse than the "*Feature-Dimension weighted-Decay RL model*". We describe the model comparison in the main text and in the Supplementary Methods.

Please also see our reply to comment 3 of reviewer 2 for describing why non-relevant feature values are low but non-zero.

Comment 3: [The reviewer felt that the results remain difficult to understand and asked for clarification of the core findings.]

Response: We regret that the reviewer conveys this opinion. We have discussed our findings with researchers from diverse fields spanning the attention, learning, cognitive control, and computational fields. Our general impression is that our four key findings are considered clear and a true progress for each of the fields mentioned. Four main findings that we highlight in the abstract and validate with statistical tests are:

1. Neurons in fronto-striatal circuits encode reward prediction errors for all three features of an attended stimulus with the most ubiquitous encoding evident for the reward-relevant feature. We were able to find this result by using a well control feature-based attention and learning task.
2. Neuronal activity indicating feature-specific prediction error encoding emerge on average later than general (non-selective) prediction error signaling. We were able to find this result by recording not from a single brain area but by four brain areas at the same time.
3. The ACC sticks out among the four different brain areas by showing on average the earliest feature-specific prediction error encoding neuronal firing rate modulations. We were able to find this result by recording not from a single brain area but by four brain areas at the same time.
4. The feature-specific encoding of prediction error signaling is directly linked to enhanced firing to the predictive cue (feature onset of the stimuli) in later trials. We were able to find this result by using a reversal learning task that allowed multiple repetitions of learning (reversals) and exploitation periods (later trials in the reversal learning block).

These findings emerge from a diverse set of complex findings that are expected for multi-area recordings.

We sometimes have the impression that it would be easier to record data and interpret findings in a single brain area only. For our study, it is a novel finding to report feature based learning effects in any of the recorded brain areas. For example, we recorded feature-specific surprise signals in the ventral striatum in a feature-based attention task, which is an unprecedented insight into how attention biases might be linked to motivational and teaching signals. However, we believe that describing this ventral striatum signature of feature-based learning in the larger network has a critical benefit. In our case, we can single out the ACC as providing on average similar signals at an earlier time.

Reviewer #2 (Remarks to the Author):

Comment general: The authors have addressed most of my concerns. However, I have a few remaining comments.

Response: We are glad that we could address most of the points.

Comment 1: With respect to this statement: "Please note that RPE is computed as $RPE = R - V$ (where R denotes reward, 1 or 0, and V denotes expected value of the chosen stimulus), indicating that RPE at the time of reward is anti-correlated with the value of the chosen stimulus prior to the time of reward, making RPE and 'chosen-value' non-independent variables that are difficult to distinguish."

This will not be true during reversal learning. Because values change so they are not always 0/1. This should be clarified and addressed.

Response: We believe there has been a misunderstanding. While values are not always 0/1, rewards are always 0/1 and that doesn't change with reversal learning. Thus, with rewards being 0/1 our statement "...RPE at the time of reward is anti-correlated with the value of the chosen stimulus prior to the time of reward..." is true.

More specifically, the statement we wrote translates the preceding mathematical formula in words and therefore also holds during reversal learning. Correlation is mathematically defined as the average of the product of two variables, corrected for their means and normalized such that when two variables are a (positive) multiple of each other it is unity. Hence, for a relation $RPE = -V + \text{noise}$, the correlation is -1 in the absence of noise, and decreases towards zero when the noise becomes larger and larger. Hence, the two variables always remain anticorrelated.

In the formula in question, $RPE = R - V$, the RPE is thus anticorrelated with V and correlated with R . In the paper, we correlate the firing rate of neurons with the RPE in order to find neurons that have a significant correlation with RPE. The same neurons have, out of mathematical necessity also a (negative) correlation with V . This is particularly relevant because we split analyses into negative (R always 0) and positive RPE (R always 1) encoding and specifically partial out R for the analysis of surprise encoding. Based on the presence of a correlation, the two possibilities (correlate with V versus correlate with RPE) cannot be easily distinguished, except when looking at a time that precedes the reward, in which case there is not yet an R . We have clarified the statement in the text by incorporating this reasoning more explicitly to avoid potential confusion. We write in the revised manuscript:

"Please note that RPE is computed as $RPE = R - V$, where R denotes reward, (always 1 or 0), and V denotes expected value of the chosen stimulus. This formulation indicates that RPE is positively correlated with reward (R , always 0/1), and anti-correlated ($-V$) with the value of the chosen stimulus prior to the time of reward. The anticorrelation of RPE at the time of reward and the value of the chosen stimulus prior to the time of reward is constant throughout reversal learning because reward is always 0 and 1." (p. 8)

Comment 2: The Bayesian model was confusing. I was expecting that the authors would use a binomial distribution to represent the reward probabilities associated with each color. Bayesian models are problematic with deterministic outcomes, because there is no way to model the

deterministic mappings. I could not really sort out what f was in the development of this model? How was p_r calculated?

Response: We acknowledge the question very much and are grateful to add more clarity to the Bayesian model.

The Bayesian model recurrently updates posterior probabilities from prior information, which then become new priors for the subsequent trial. For the estimate of what the target feature is, the algorithm starts from uniform priors. Then, trial after trial, posterior probabilities are updated according to the predictability of reward. Thus, the posterior probability of a feature only survives (i.e., it is different from zero) while that feature is associated with the relevant object. After very few trials, this Bayesian approach is able to identify the specific stimulus feature that is deterministically predictive of reward. On reversals (i.e., at each block transition), all posteriors become zero, at which point, the algorithm resets to uniform probabilities.

The pure Bayesian model is not a good fit of monkeys' behavior (it learns too fast compared to animals). However, using the Bayesian learning to determine the relevant stimulus dimension, and weight reinforcement learning values accordingly, provided the best fit against alternative models (illustrated in **Fig S1b,c**; see equations 4-9 in Supplementary Methods; see also Hassani et al., Sci Rep 2017). Such a Bayesian-RL model was first proposed by Niv and colleagues (Wilson et al., Front Hum Neurosci 2012; Niv et al., J Neurosci 2015).

In summary, Bayesian models can be used with deterministic reward as one still needs to learn what feature gave rise to the reward. In our task, if an object is rewarded, there are three feature-value combinations that could be the target.

Comment 3: I understand that the stimuli the animals were choosing did in fact have additional features other than color. But the values of these features should never go above 0 on average. So I am still not clear why the authors think that there will be any feature specific prediction errors about these dimensions. This was not clarified in the introduction. The task used does not allow the authors to address these issues, unless it can be shown that the animals did tend to try to use these other dimensions for learning. But then this raises the question of why they would do so, since those dimensions did not predict reward.

Response: We like to reply with four arguments and added clarifications accordingly to the revised manuscript.

First, motion and location were only randomly correlated with reward, but in each single trial both of these features were behaviorally relevant and attended and directly instrumental to receive reward: Subjects were required to make a saccade in the direction of motion of the spatially attended stimulus to receive reward. This aspect of the task made it particularly difficult for the animals (and for humans trying this task) as evidenced by an overall asymptotic performance of 'only' 75% rewarded choices. We can thus expect that the credit assignment of the outcomes is affected by motion and location information, despite the fact that these variables were only on a single trial level, but not systematically, linked to reward. To describe this aspect explicitly we write, e.g., in the results section of the revised manuscript:

"This task required (1) feature-based attentional selection of one over another stimulus based on a reward associated color, (2) to use the motion direction of the attended

stimulus to program a saccadic response, and (3) to make a response only when the attended stimulus dimmed. Therefore, stimulus location (spatial attention) and stimulus motion-direction were relevant to the response on a trial-by-trial basis, while only stimulus color was linked to reward across trials.” (p. 5)

Second, our model comparison addresses the reviewer’s point directly: The behavior of the monkeys was best explained by a model that learns the feature value of the most reward predictive feature (color), but that incorporates values from the stimulus motion and location. This “*Feature-Dimension weighted-Decay RL model*” was better than a model that only has color as relevant task features without representing motion and location at all (our *Feature-selective RL model*). The feature-dimension-weighted model is also better than a model that treats all features similarly without weighting (our *Feature-Nonselective RL model*). Therefore, the data suggest that values of motion and location do affect learning. With the weighting of feature dimensions, the influence of motion and location features very quickly reduces to near zero values within few trials after the block reversal. This finding can be seen in the new Supplementary **Figure S13** that we added in response to Comment 1 of Reviewer 1, in which we calculated the prediction error for the values of each of the chosen stimulus’ features separately ($RPE_{\text{feature}} = \text{Outcome} - V_{\text{feature}}$). This figure allows seeing that values for the motion and location features are not zero. If they would be zero, the positive RPE_{location} and $RPE_{\text{motion direction}}$ after correct trials (Outcome is 1) would be 1, and the negative RPE_{location} and $RPE_{\text{motion direction}}$ after error trials (Outcome is 0) would be 0. Instead **Fig S13A,C** shows that positive RPEs fluctuate around 0.5, and **Fig S13B,D** shows that negative RPEs fluctuate around -0.1. This illustrates that noisy fluctuations of model-calculated values for the motion direction and location features exist. Because this model was superior to the model that only considered color, we believe this provides the requested evidence that non-relevant feature dimensions have subtle influences on behavior.

Third, the reviewer conveys that “*value of these [motion and location] features should never go above 0 on average*”. This cannot be correct because the values are in [0, 1] and it is thus impossible that their mean value remains at 0 (even though one may indeed expect that their noisy value is low).

Fourth, regarding the question “why they [the animals] would do so [track feature specific prediction errors about non-relevant dimensions], since those dimensions did not predict reward”, please note that the animals are not informed about relevance in any other way than by a global binary reward feedback signal. In order for the animals to arrive at this very same conclusion (that those other features are not relevant), they need to track feature value fluctuations in correlation with reward delivery updated by feature specific RPEs.

In summary, the Bayesian weighting of feature dimensions provides a central mechanism to reduce quickly the interference from reward-irrelevant feature dimensions that are only spuriously, but not causally, linked to the reward outcome. Once these features are identified as non-relevant by Bayesian inference, those values have near-zero weighting, and no interference is propagated to the final decision.

This mechanism is likely of particular behavioral importance, because when reward contingencies change (after a reversal) the subject is able to retain flexibility when it can track the RPE from all features that could be causing reward outcome but increases reward intake when it can quickly prevent interference from reward irrelevant features using e.g. such a Bayesian weighting mechanism.

The Bayesian weighting provides a parsimonious way for a model to learn relevant feature dimensions – similar to re-configuring an attentional set - without the need of an experimenter to explicitly specify what is relevant in respect of the subjective experience of the experimental subjects.

Reviewer #3 (Remarks to the Author):

Comment 1: The authors have made many improvements in the manuscript that have greatly clarified their findings. The overall conceptual design of the experiment and the meaning of the findings is now clear. I have no further questions. I think this is an interesting contribution and deserves publication in Nat. Communication.

Response: Thank you.

REVIEWERS' COMMENTS:

Reviewer #2 (Remarks to the Author):

I have no further comments.